# MORA: ENABLING GENERALIST VIDEO GENERATION VIA A MULTI-AGENT FRAMEWORK

## ABSTRACT

Text-to-video generation has made significant strides, but replicating the capabilities of advanced systems like OpenAI's Sora remains challenging due to their closed-source nature. Existing open-source methods struggle to achieve comparable performance, often hindered by ineffective agent collaboration and inadequate training data quality. In this paper, we introduce Mora, a novel multi-agent framework that leverages existing open-source modules to replicate Sora's functionalities. We address these fundamental limitations by proposing three key techniques: (1) multi-agent fine-tuning with a self-modulation factor to enhance inter-agent coordination, (2) a data-free training strategy that uses large models to synthesize training data, and (3) a human-in-the-loop mechanism combined with multimodal large language models for data filtering to ensure high-quality training datasets. Our comprehensive experiments on six video generation tasks demonstrate that Mora achieves performance comparable to Sora on VBench (Huang et al., 2024), outperforming existing open-source methods across various tasks. Specifically, in the text-to-video generation task, Mora achieved a Video Quality score of 0.800, surpassing Sora's 0.797 and outperforming all other baseline models across six key metrics. Additionally, in the image-to-video generation task, Mora achieved a perfect Dynamic Degree score of 1.00, demonstrating exceptional capability in enhancing motion realism and achieving higher Imaging Quality than Sora. These results highlight the potential of collaborative multi-agent systems and human-in-the-loop mechanisms in advancing text-to-video generation.

## 1 INTRODUCTION

Generative AI technologies have significantly transformed various industries, with substantial advancements particularly notable in visual AI through image generation models like Midjourney (Midjourney, 2023), Stable Diffusion 3 (Esser et al., 2024), and DALL-E 3 (Betker et al., 2023). These models have demonstrated remarkable capabilities in generating high-quality images from textual descriptions. However, progress in text-to-video generation, especially for videos exceeding 10 seconds, has not kept pace. Recent developments such as Pika (pik) and Gen-3 (Gen, b) have shown potential but are limited to producing short video clips.

A major breakthrough occurred with OpenAI's release of Sora in February 2024—a text-to-video model capable of generating minute-long videos that closely align with textual prompts. Sora excels in various video tasks, including editing, extending footage, offering multi-view perspectives, and adhering closely to user instructions (OpenAI, 2024a). Despite its impressive capabilities, Sora remains a closed-source system, which poses significant barriers to academic research and development. Its black-box nature hinders the community's ability to study, replicate, and build upon its functionalities, thereby slowing progress in the field. Attempts to reverse-engineer Sora are exploring potential techniques like diffusion transformers and spatial patch strategies (Sohl-Dickstein et al., 2015; Ho et al., 2020; Bao et al., 2023; Peebles & Xie, 2023), but a large gap still exists due to the intensive computation required to train everything from scratch with a single model.

To address these challenges, we propose Mora, a novel multi-agent framework that leverages ideas from standardized operating procedures (SOPs) and employs multiple agents using existing open-source modules to replicate the complex functionalities of Sora. While SOPs and multi-agent systems have been utilized in text-based tasks (Hong et al., 2023), applying them to text-to-video generation

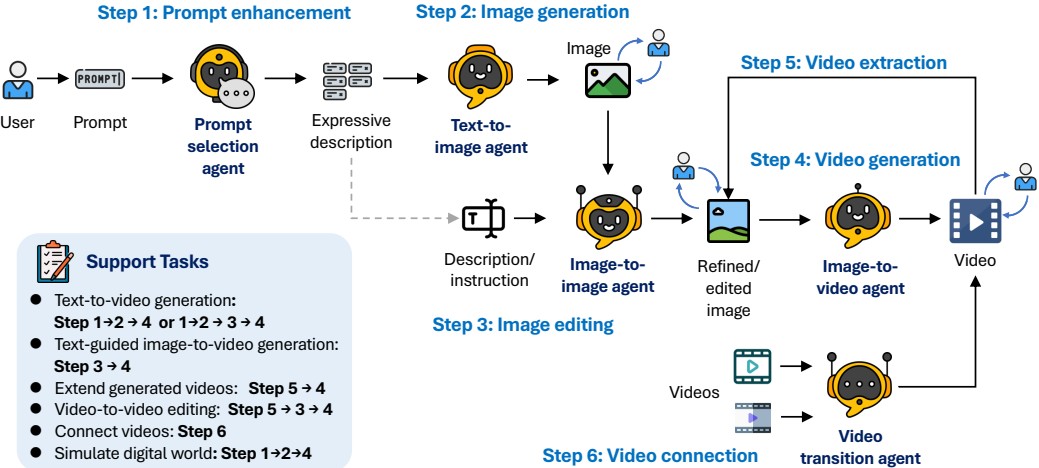

Figure 1: Illustration of SOPs to conduct video-related tasks in Mora.

presents unique challenges. Naive multi-agent frameworks (Yuan et al., 2024; Xie et al., 2024) often fail because agents lack effective collaboration mechanisms, leading to suboptimal performance. Moreover, existing multi-agent video generation approaches struggle to balance pipeline automation with the need for high-quality training data, which is critical for producing high-fidelity videos.

Collecting high-quality video data for training is time-consuming and computationally expensive (Chen et al., 2024a), and the scarcity of such data hampers the performance of open-source models. Furthermore, the quality of available datasets varies widely, making it challenging to train models that can match the performance of proprietary systems like Sora. To overcome these challenges, leveraging human-in-the-loop mechanisms for data filtering becomes essential. By integrating human expertise in the data synthesis process, we can ensure that the training data is of high quality, which is crucial for training effective video generation models.

In this paper, we address these fundamental limitations by introducing several key techniques in Mora. First, we develop a multi-agent fine-tuning approach with a novel self-modulation factor that enhances coordination among agents, allowing them to collaborate effectively to achieve common goals. Second, we employ a data-free training strategy that uses large models to synthesize training data, reducing reliance on large labeled datasets and enabling efficient model training without extensive data collection efforts. Third, we leverage human-in-the-loop mechanisms, in combination with multimodal large language models, for data filtering during the training data synthesis process. This approach ensures the quality of the synthesized training data, leading to improved performance of the video generation pipeline. We summarize the overall pipeline and supported tasks in Figure 1.

By integrating human-in-the-loop mechanisms with multimodal large language models for data filtering in our data-free training strategy, we significantly enhance the quality of the training data and, consequently, the generated videos. These techniques collectively improve inter-agent and agent-human collaboration, enhance the quality and diversity of the generated videos, and expand the system's capabilities to match those of Sora. Our comprehensive experiments on six video generation tasks demonstrate that Mora achieves performance comparable to Sora on VBench (Huang et al., 2024), closely approaching Sora and outperforming existing open-source methods across various tasks. Specifically, in the text-to-video generation, Mora achieves a Video Quality score of 0.800, surpassing Sora's 0.797, and outperforms all other baselines. In the Image-to-Video Generation task, Mora matches Sora in Video-Text Integration (0.90) and Motion Smoothness (0.98 vs. 0.99), and surpasses Sora in Imaging Quality (0.67 vs. 0.63) and Dynamic Degree (1.00 vs. 0.75). Additionally, in Video-to-Video Editing, Mora matches Sora in both Imaging Quality and Temporal Style, each scoring 0.52 and 0.24, respectively. These results demonstrate Mora's ability to not only replicate but also enhance the functionalities of Sora, providing a promising platform for future research.

We summarize our main contributions as follows:

- We introduce Mora, a novel multi-agent framework that leverages existing open-source modules to replicate the functionalities of Sora in text-to-video generation.

- We address fundamental challenges in agent collaboration for video generation through three key designs: (1) a self-modulated multi-agent fine-tuning approach with dynamic modulation factor for enhanced coordination, (2) human-in-the-loop control mechanisms enabling real-time adjustments, and (3) a data-free training strategy using large models to synthesize diverse training data. These techniques significantly improve inter-agent cooperation and output quality in multi-agent video generation systems.
- We demonstrate through extensive experiments that Mora achieves performance comparable to Sora on standard benchmarks, advancing the field of text-to-video generation and providing an accessible platform for future research.

## 2 RELATED WORK

### 2.1 TEXT-TO-VIDEO GENERATION

Generating videos based on textual descriptions has been a long-discussed topic. While early efforts in the field were primarily rooted in GANs (Wang et al., 2020; Chu et al., 2020) and VQ-VAE (Yan et al., 2021), recent breakthroughs in generative video models, driven by foundational work in transformer-based architectures and diffusion models, have significantly advanced academic research. Auto-regressive transformers are early leveraged in video generation (Wu et al., 2022; Hong et al., 2022b; Kondratyuk et al., 2023). These models are designed to generate video sequences frame by frame, predicting each new frame based on the previously generated frames. Parallelly, the adaptation of masked language models (He et al., 2021) for visual contexts, as demonstrated by (Gupta et al., 2022; Villegas et al., 2022; Yu et al., 2023; Gupta et al., 2023), underscores the versatility of transformers in video generation. The recently proposed VideoPoet (Kondratyuk et al., 2023) leverages an auto-regressive language model and can multitask on a variety of video-centric inputs and outputs.

In another line, large-scale diffusion models (Sohl-Dickstein et al., 2015; Ho et al., 2020) show competitive performance in video generation (Ho et al., 2022a; Singer et al., 2022; Khachatryan et al., 2023; Wu et al., 2023a; Du et al., 2024). By learning to gradually denoise a sample from a normal distribution, diffusion models (Sohl-Dickstein et al., 2015; Ho et al., 2020) implement an iterative refinement process for video synthesis. Initially developed for image generation (Rombach et al., 2022c; Podell et al., 2023), they have been adapted and extended to handle the complexities of video data. This adaptation began with extending image generation principles to video (Ho et al., 2022b;a; Singer et al., 2022), by using a 3D U-Net structure instead of conventional image diffusion U-Net. In the follow-up, latent diffusion models (LDMs) (Rombach et al., 2022b) are integrated into video generation (Zhou et al., 2022; Chen et al., 2023b; Wang et al., 2023a; Chen et al., 2024a), showcasing enhanced capabilities to capture the nuanced dynamics of video content. For instance, Stable Video Diffusion (Blattmann et al., 2023) can conduct multi-view synthesis from a single image while Emu Video (Girdhar et al., 2023) uses just two diffusion models to generate higher-resolution videos.

Researchers have delved into the potential of diffusion models for a variety of video manipulation tasks. Notably, Dreamix (Molad et al., 2023) and MagicEdit (Liew et al., 2023) have been introduced for general video editing, utilizing large-scale video-text datasets. Conversely, other models employ pre-trained models for video editing tasks in a zero-shot manner (Ceylan et al., 2023; Couairon et al., 2023; Yang et al., 2023; Chai et al., 2023). SEINE (Chen et al., 2023c) is specially designed for generative transition between scenes and video prediction. The introduction of diffusion transformers (Peebles & Xie, 2023; Bao et al., 2023; Ma et al., 2024a) further revolutionized video generation, culminating in advanced solutions like Latte (Ma et al., 2024b) and Sora (OpenAI, 2024a). There is also utilization in a specific domain such as Bora (Sun et al., 2024b) in biomedical scenarios. Sora's ability to produce minute-long videos of high visual quality that faithfully follow human instructions heralds a new era in video generation.

### 2.2 AI AGENTS

Large models have enabled agents to excel across a broad spectrum of applications, showcasing their versatility and effectiveness. They have greatly advanced collaborative multi-agent structures for multimodal tasks in areas such as scientific research (Tang et al., 2024), software development (Hong et al., 2023; Qian et al., 2023) and society simulation (Park et al., 2023). Compared to individual

agents, the collaboration of multiple autonomous agents, each equipped with unique strategies and behaviors and engaged in communication with one another, can tackle more dynamic and complex tasks (Guo et al., 2024). Through a cooperative agent framework known as role-playing, Li et al. (2024b) enables agents to collaborate and solve complex tasks effectively. Park et al. (2023) designed a community of 25 generative agents capable of planning, communicating, and forming connections. Liang et al. (2023) have explored the use of multi-agent debates for translation and arithmetic problems, encouraging divergent thinking in large language models. Hong et al. (2023) introduced MetaGPT, which utilizes an assembly line paradigm to assign diverse roles to various agents. In this way, complex tasks can be broken down into subtasks, which makes it easy for many agents working together to complete. Xu et al. (2023) used a multi-agent collaboration strategy to simulate the academic peer review process. Besides, AutoGen (Wu et al., 2023b) is a generic programming framework that can be used to implement diverse multi-agent applications across different domains, using a variety of agents and conversation patterns. This motivates our focus on applying multi-agent frameworks on text-to-video generation tasks, enabling agents to collaborate seamlessly from project inception to completion.

## 3 MORA: A MULTI-AGENT FRAMEWORK FOR VIDEO GENERATION

Current text-to-video generation models directly generate videos from textual inputs, which prevents users from supervising key aspects of video quality, style, and other important elements in real-time. To address this limitation, we propose a novel multi-agent system coupled with a self-modulated training algorithm specifically designed for the video generation task. In the subsequent sections, we outline our approach in detail. Section 3.1 describes the problem and design of our multi-agent system and the architecture of our model. Finally, in Section 3.2 and Appendix A.7, we present our data-free multi-agent fine-tuning method.

### 3.1 AGENT ARCHITECTURE OF MORA

In this section, we introduce the problem and our multi-agent video generation system. To address the complexity of generating high-quality, long-duration videos, we propose a multi-agent framework where the generative model $G$ is composed of $n$ collaborating agents $\{A_1, A_2, \cdots, A_n\}$. As shown in Figure 1, each agent specializes in a specific subtask and collaborates together within the video generation pipeline. We further introduce the definition of each agent below.

**Problem Definition** Let $P \in \mathcal{P}$ denote a textual prompt from the space of all possible prompts $\mathcal{P}$, describing the desired video content. A video $V$ is represented as a sequence of frames $V = \{F_1, F_2, \cdots, F_T\}$, where each $F_t \in \mathbb{R}^{H \times W \times C}$ corresponds to an image at time step $t$, with height $H$, width $W$, and $C$ color channels. Our goal is to generate extended-length videos that are semantically consistent with the textual prompt while exhibiting high visual quality and temporal coherence. Formally, we aim to learn a generative model $G : \mathcal{P} \to \mathcal{V}$ that maps a textual prompt $P$ to a video $V$ in the space of all possible videos $\mathcal{V}$: $V = G(P)$. The quality of the generated video can be assessed using a set of metrics $\mathcal{M} = \{m_1, m_2, ..., m_K\}$, where each $m_i : \mathcal{V} \times \mathcal{P} \to \mathbb{R}$ evaluates a specific aspect of the video (e.g., visual quality, temporal consistency, semantic alignment with the prompt). Our objective is to maximize these quality metrics while ensuring diversity in the generated videos. In our multi-agent framework, $G$ is decomposed into a set of specialized agents $\{A_1, A_2, ..., A_N\}$, each responsible for a specific subtask in the video generation process. These agents collaborate to produce the final video output, allowing for more granular control of the generation process.

**Definition and Specialization of Agents.** The specialization of agents enables flexibility in the breakdown of complex work into smaller and more focused tasks, as depicted in Figure 1. In our framework, we have five agents: prompt selection and generation agent $A_1$, text-to-image generation agent $A_2$, image-to-image generation agent $A_3$, image-to-video generation agent $A_4$, and video-to-video agent $A_5$. We present brief descriptions below, and detailed definitions can be found in Appendix A.5.

- Prompt Selection and Generation Agent ($A_1$): This agent employs large language models like GPT-4 and Llama (Achiam et al., 2023; Touvron et al., 2023) to analyze and enhance textual prompts, extracting key information and actions to optimize image relevance and quality.

- Text-to-Image Generation Agent ($A_2$): Utilizing models such as those by Rombach et al. (2022c) and Podell et al. (2023), this agent translates enriched textual descriptions into high-quality images by deeply understanding and visualizing complex inputs.
- Image-to-Image Generation Agent ($A_3$): Referencing Brooks et al. (2023), this agent modifies source images based on detailed textual instructions, accurately interpreting and applying these to make visual alterations ranging from subtle to transformative.
- Image-to-Video Generation Agent ($A_4$): As described by Blattmann et al. (2023), this model transitions static images into dynamic video sequences by analyzing content and style, ensuring temporal stability and visual consistency.
- Video Connection Agent ($A_5$): This agent creates seamless transition videos from two input videos by leveraging keyframes and identifying common elements and styles, ensuring coherent and visually appealing outputs.

**General Structure.** Mora model structure is depicted in Figure 1. Specifically, given a user input $T$, the Prompt Enhancement Agent ($A_1$) first refines $T$ into a form better suited for video generation. The enhanced prompt is then passed to the Text-to-Image Agent ($A_2$) to generate the first frame of the video. At this stage, the user can review and confirm whether the tone and quality of the frame meet their expectations. If not, the user can either request a re-generation or pass the frame to the Image-to-Image Agent ($A_3$) for adjustments. Once the desired first frame is finalized, it is forwarded to the Image-to-Video Agent ($A_4$) to generate a high-quality video that aligns with the user's requirements. This step-by-step, independently controllable, and interactive process ensures that the final video more closely meets the user's expectations while maintaining high quality. In cases where a user wishes to generate a continuous video from different video clips, $A_5$ analyzes the final frame of the preceding video and the initial frame of the next and ensures a smooth blending between them. Moreover, this procedural design ensures that the generation process does not have to start from scratch, and the human-in-the-loop technique makes the entire pipeline more controllable, as detailed in Appendix A.4. It enables our framework to handle various tasks, such as image-to-video generation, and even video extension and stitching. For more details about supported tasks, please refer to Appendix A.3. In addition to prompt-based generation, Mora structure also supports task-wise model fine-tuning, ensuring that agents can effectively follow instructions and consistently produce high-quality content.

## 3.2 SELF-MODULATED MULTI-AGENT FINETUNING

In this section, we introduce our proposed multi-agent finetuning design, based on the previously described model structure. Directly prompting each agent does not account for the downward transmission of information, which could lead to inefficiencies or errors in communication between agents. Additionally, the impact of each agent on the final result is not uniform. To address these issues, we adopt an end-to-end training approach. Our proposed training procedure involves (1) a self-modulation factor to enhance inter-agent coordination, (2) a data-free training strategy to synthesize training data, (3) LLM selection with human-in-the-loop to control the training data quality. In the following sections, we provide a detailed explanation of each component.

**Self-modulated Fine-tuning Algorithm.** Previous methods primarily employ direct end-to-end fine-tuning across the entire task procedure, while others may fine-tune individual agents based on specific tasks. However, both approaches can influence model performance: (1) end-to-end fine-tuning may result in improper loss allocation for each agent, and (2) fine-tuning agents individually can lead to misaligned distributions. To address the limitations of existing fine-tuning approaches, we propose a self-modulated fine-tuning algorithm specifically designed for our multi-agent model structure. This method aims to enhance coordination among agents and improve overall performance by introducing a modulation embedding that dynamically adjusts the influence of each agent during the generation process. The key motivations behind our approach are to balance the impact of each agent on the final output, improve inter-agent communication and coordination, and allow for dynamic adjustment of agent contributions based on the current task and intermediate outputs.

Our design introduces a modulation embedding that is concatenated with the text embedding of the enhanced prompt, allowing for fine-grained control over the generation process. This embedding is optimized alongside the model parameters during training, enabling the system to learn optimal

---

**Algorithm 1:** Self-Modulated Multi-Agent Video Generation with Data-free Fine-tuning

---

**Input:** Initial agents' parameters $\{\theta_i\}$; initial modulation embeddings $\{\mathbf{z}_i = \mathcal{E}_i(\texttt{"[Mod]"})\}_{i=1}^n$; number of iterations $N$; number of samples per iteration $S$; prompt set $P$; batch size $B$; number of epochs per iteration $K$; learning rates $\eta_{\theta_i}, \eta_{z_i}$

**Output:** Trained model parameters $\{\theta_i\}$

**1 for** *iteration* $n = 1$ **to** $N$ **do**

**2**    Construct training dataset $\mathcal{D}_n$ using the data-free training strategy (see Sec. 3.2 );

**3**    **for** *epoch* $e = 1$ **to** $K$ **do**

**4**      **foreach** *batch* $\{(P^{(b)}, V_{target}^{(b)})\}_{b=1}^B$ *in* $\mathcal{D}_n$ **do**

**5**        Compute text embeddings $\mathbf{e}_i^{(b)} = \mathcal{E}_i(P^{(b)})$;

**6**        **for** $i = 1$ **to** $n$ **do**

**7**          Concatenate agent-specific modulation embedding with text embedding for all examples: $\tilde{\mathbf{e}}_i^{(b)} = [\mathbf{e}_i^{(b)}; \mathbf{z}_i]$;

**8**          Generate outputs for the batch: $O_i^{(b)} = \mathcal{M}_i(O_{i-1}^{(b)}, \tilde{\mathbf{e}}_i^{(b)})$, where $O_0^{(b)} = \emptyset$;

**9**        Compute the final loss for the batch: $L_{\text{final}} = \frac{1}{B} \sum_{b=1}^B L(O_n^{(b)}, V_{\text{target}}^{(b)})$;

**10**        **for** $i = 1$ **to** $n$ **do**

**11**          Update agent-specific modulation embedding: $\mathbf{z}_i \leftarrow \mathbf{z}_i - \eta_{z_i} \frac{\partial L_{\text{final}}}{\partial \mathbf{z}_i}$;

**12**          Compute modulation factor: $||\mathbf{z}_i||_2 = \sqrt{\sum_k z_{i,k}^2}$;

**13**          Compute modulation factor for model $i$: $\alpha_i = ||\mathbf{z}_i||_2/n$;

**14**          Update model parameters with dynamic learning rate: $\theta_i \leftarrow \theta_i - \alpha_i \eta_{\theta_i} \frac{\partial L_{\text{final}}}{\partial \theta_i}$;

**15 return** $\{\theta_i\}$;

---

coordination strategies. By doing so, we ensure that each agent can adapt its output based on the state of the preceding agent, leading to more coherent and high-quality video generation.

The implementation of our self-modulated fine-tuning algorithm begins with the initialization of a modulation embedding $\{\mathbf{z}_i\}$ using the text embedding of a special token (Ning et al., 2023): $\{\mathbf{z}_i = \mathcal{E}_i(\texttt{"[Mod]"})\}$, where $\mathcal{E}_i$ is the text encoder of agent $i$. For each enhanced prompt $P_{\text{enh}}$, we compute its text embedding $\{\mathbf{e}_i = \mathcal{E}_i(P_{\text{enh}})\}$ and concatenate it with the modulation embedding: $\{\tilde{\mathbf{e}}_i = [\mathbf{e}_i; \mathbf{z}_i]\}$. This combined embedding serves as input to the agents.

Each agent $\mathcal{M}_i$ in the system processes its input and produces an output $O_i$. For the first agent, $O_1 = \mathcal{M}_1(\tilde{\mathbf{e}})$, and for subsequent agents ($i > 1$), $O_i = \mathcal{M}_i(O_{i-1}, \tilde{\mathbf{e}})$. The modulation factor, which influences the impact of each agent, is calculated as the L2 norm of $\mathbf{z}_i$: $||\mathbf{z}_i||_2 = \sqrt{\sum_k z_k^2}$.

During training, we minimize the total loss $L_{\text{total}} = L(O_n, V_{\text{target}})$ between the final output $O_n$ and the target video $V_{\text{target}}$. Both the model parameters $\theta_i$ of each agent and the modulation embedding $\mathbf{z}_i$ are updated using gradient descent: $\theta_i \leftarrow \theta_i - \eta_{\theta_i} \frac{\partial L_{\text{total}}}{\partial \theta_i}$ and $\mathbf{z}_i \leftarrow \mathbf{z}_i - \eta_z \frac{\partial L_{\text{total}}}{\partial \mathbf{z}_i}$. By optimizing $\mathbf{z}$, the modulation embedding learns to adjust its influence to minimize the loss, effectively enhancing inter-agent coordination and ensuring that each agent can dynamically adjust its output according to the state of the preceding agent.

The complete training process, including the initialization of the modulation embedding, the forward pass through the agents, and the optimization of both model parameters and the modulation embedding, is detailed in Algorithm 1. This algorithm encapsulates our self-modulated fine-tuning approach, providing a comprehensive framework for improving the performance and coordination of our multi-agent video generation system.

**Multimodal LLM Selection with Human-in-the-loop Control.** Despite the availability of numerous open-source video datasets, their quality varies significantly, making it challenging for the pretrained agents we use to effectively leverage these datasets to improve video generation performance. Also, manually filtering high-quality data is time-consuming and inefficient. To address this issue, we introduce a multimodal LLM data selection procedure with human-in-the-loop control.

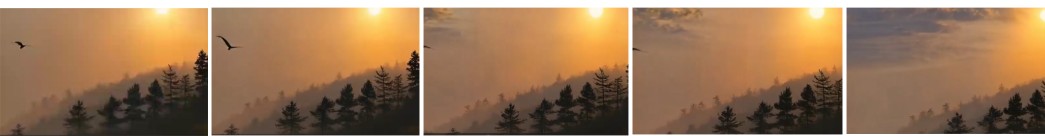

*A serene sunrise over a misty forest: as the sun rises and mist shrouds the treetops, birds glide gracefully across the sky above partially obscured hills.*

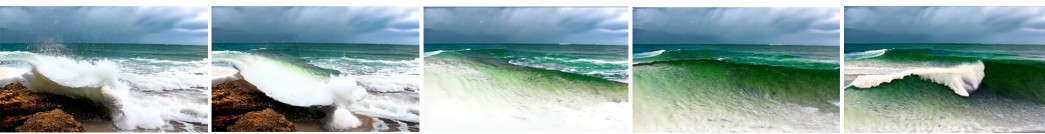

*Waves crash onto a rocky beach under a partly cloudy sky, hitting jagged rocks with force and sending up sprays of white foam before retreating back into the sea.*

Figure 2: Samples for text-to-video generation of Mora. Our approach can generate high-resolution, temporally consistent videos from text prompts. The samples shown are 480p resolution over 12 seconds duration at 276 frames in total.

We first sample a batch of candidates from the agent system. Based on the generated candidates, we further leverage strong multimodal LLMs that support multi-frame and multi-video inputs to provide evaluation for the candidate set. When multiple multimodal LLMs agree on the best video from a set, we directly include it in the training dataset, the evaluation prompt examples detailed in Appendix A.6. However, when the LLMs' evaluations differ, we introduce human reviewers for secondary screening. In such cases, human reviewers evaluate the generated videos and select the one with the highest quality for inclusion in the training set. If the reviewers determine that none of the candidates meet the quality standard, the entire set is discarded. This approach enhances the stability and robustness of the filtering process by combining LLMs' automated assessments with human judgment, ensuring the model can handle complex or ambiguous cases.

**Data-free Training Strategy.** During training, instead of using open-source datasets directly, we adopt a data-free strategy to synthesize training data dynamically. This method addresses challenges like inconsistent quality in open-source datasets and the lack of intermediate outputs required for training different agents. Given a user input prompt set $P = \{P^{(1)}, P^{(2)}, \ldots, P^{(S)}\}$, we first utilize the LLM to generate an enhanced set of diverse prompts $P_{\text{enh}} = \{P_{\text{enh}}^{(1)}, P_{\text{enh}}^{(2)}, \ldots, P_{\text{enh}}^{(S)}\}$. For each enhanced prompt $P_{\text{enh}}^{(s)}$, our initial workflow—comprising non-finetuned models with parameters $\{\theta_i\}_{i=1}^{M}$—is applied to generate a set of candidate videos $\mathcal{C}_s = \{V_1, V_2, V_3, V_4\}$. Next, we use a human-in-the-loop process, in combination with multimodal LLM selection, to select the highest quality video $\hat{V}^{(s)}$ from the candidate set $\mathcal{C}_s$ for each prompt $P_{\text{enh}}^{(s)}$. The selected video $\hat{V}^{(s)}$ and corresponding prompt $P_{\text{enh}}^{(s)}$ form the training dataset for iteration $n$, denoted by $\mathcal{D}_n = \{(P_{\text{enh}}^{(s)}, \hat{V}^{(s)})\}_{s=1}^{S}$. Using the constructed dataset $\mathcal{D}_n$, self-modulated multi-agent fine-tuning is performed to update the agent parameters $\{\theta_i\}_{i=1}^{M}$, where $M$ is the number of agents involved. The newly fine-tuned model parameters $\{\theta_i\}_{i=1}^{M}$ then become the starting point for the next round of data generation and fine-tuning, enabling iterative improvement over $N$ iterations. Thus, after $N$ iterations, the final optimized agent parameters $\{\theta_i\}$ are obtained.

## 4 EXPERIMENTS

### 4.1 SETUP

**Tasks.** We evaluated our proposed Mora framework on six diverse video generation tasks to comprehensively assess its capabilities. The six tasks are as follows: **(1) Text-to-Video Generation**, where videos are generated directly from textual prompts; **(2) Image-to-Video Generation**, which involves creating videos conditioned on both an input image and a textual description; **(3) Extend Generated Videos**, aiming to extend existing videos to produce longer sequences while maintaining content consistency; **(4) Video-to-Video Editing**, which edits existing videos based on textual instructions to produce modified versions; **(5) Connect Videos**, focusing on seamlessly connecting two videos to

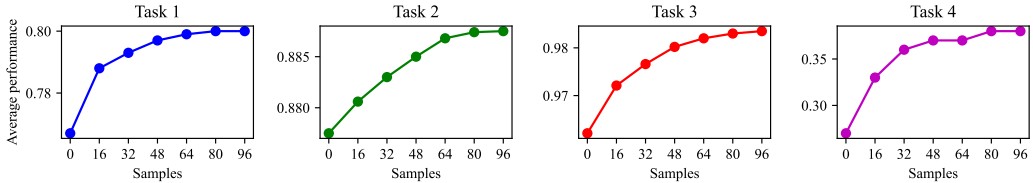

Figure 3: Performance variations of Task 1 to Task 4 across different self-training iterations.

create a longer continuous sequence; and **(6) Simulate Digital Worlds**, where videos that simulate digital or virtual environments are generated from textual prompts.

**Data.** For the text-to-video generation task, we utilized textual prompts inspired by those provided in the official Sora technical report (OpenAI, 2024b). To expand our dataset, we employed GPT-4 (OpenAI, 2023) in both few-shot and zero-shot settings to generate additional

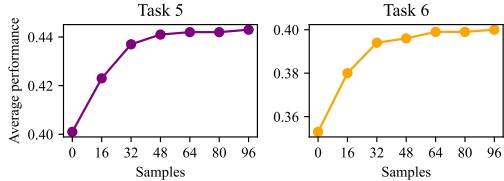

Figure 4: Performance variations of Task 5 and Task 6 across different self-training iterations.

prompts. These prompts were then used as inputs for the text-to-video models to generate videos. For the other tasks, we used relevant input data appropriate for each task, such as images, videos, or textual instructions. For comparison with Sora, we utilized videos featured on its official website and technical report.

**Baseline.** In the text-to-video generation process, we selected and compared a wide range of open-source and commercial models, including Sora (OpenAI, 2024a), VideoCrafter1 (Chen et al., 2023a), MedelScope (Mod), Show-1 (Zhang et al., 2023a), Pika (pik), LaVie and Lavie-Interpolation (Wang et al., 2023a), Gen-2 (Gen, a), and CogVideo (Hong et al., 2022a). In the other five tasks, we compare Mora with Sora. More visual comparisons and detailed analysis of the 5 tasks can be found in the Appendix A.3.

**Metrics.** We employed a combination of standard and self-defined metrics to evaluate performance across the six tasks. For text-to-video generation, we employed several metrics from Vbench (Huang et al., 2024) for comprehensive evaluation from two aspects: video quality and video condition consistency. For video quality measurement, we use six metrics: ❶ Object Consistency, ❷ Background Consistency, ❸ Motion Smoothness, ❹ Aesthetic Score, ❺ Dynamic Degree and ❻ Imaging Quality. For measuring video condition consistency, we use two metrics: ❶ Temporal Style and ❷ Appearance Style. When evaluating other tasks, due to the lack of quantitative metrics, we design the following four metrics: ❶ Video-Text Integration $VideoTI$, ❷ Temporal Consistency $TCON$, ❸ Temporal coherence $Tmean$ and ❹ Video Length. For the details of these metrics, please refer to the Appendix A.2.

### 4.2 TRAINING

In the training process, We initially used GPT-4o to generate 16 different text prompts and directly produced four videos for each prompt using an untrained model. During the data selection phase, we utilized two Multimodal LLMs, LLaVA-OneVision (Li et al., 2024a) and Qwen2-VL (Wang et al., 2024), to evaluate each set of videos and select the best-performing sample for each prompt. In the training loop, newly generated text prompts from GPT-4o were used, and after the selection process, the best videos were fed into the trained model for further optimization, and a total of 96 data were generated. Figures 3 and Figure 4 illustrate the impact of training on the performance of different tasks. It is evident that, during the early iterations, all tasks exhibit significant performance improvements, though the rate of improvement gradually slows as more iterations are completed. For details on the hyper-parameters settings in these training, please refer to Appendix A.9.

### 4.3 RESULTS

**Text-to-Video Generation.** The quantitative results are detailed in Table 1. Mora demonstrated outstanding performance in its two untrained versions, the Dynamic Degree of Mora (SVD) achieves

Table 1: Comparative analysis of text-to-video generation performance between Mora and various other models. The **Others** category scores are derived from the Hugging Face leaderboard. For **Our Mora** (SVD), categorized into three types based on the number of moving objects in the videos: Type I (single object in motion), Type II (two to three objects in motion), and Type III (more than three objects in motion). Descriptions of Mora (SVD) and Mora (Open-Sora-Plan) are detailed in Appendix A.8. $\mp$ indicates that without Self-Modulated Multi-Agent Finetuning.

| Model | Video Quality | Object Consistency | Background Consistency | Motion Smoothness | Aesthetic Quality | Dynamic Dgree | Imaging Quality | Temporal Style | Video Length(s) |
|---|---|---|---|---|---|---|---|---|---|
| **Others** | | | | | | | | | |
| Sora | 0.797 | 0.95 | 0.96 | **1.00** | 0.60 | 0.69 | 0.58 | **0.35** | **60** |
| VideoCrafter1 | 0.778 | 0.95 | **0.98** | 0.95 | 0.63 | 0.55 | 0.61 | 0.26 | 2 |
| ModelScope | 0.758 | 0.89 | 0.95 | 0.95 | 0.52 | 0.66 | 0.58 | 0.25 | 2 |
| Show-1 | 0.751 | 0.95 | 0.98 | 0.98 | 0.57 | 0.44 | 0.59 | 0.25 | 3 |
| Pika | 0.741 | 0.96 | 0.96 | 0.99 | 0.63 | 0.37 | 0.54 | 0.24 | 3 |
| LaVie-Interpolation | 0.741 | 0.92 | 0.97 | 0.97 | 0.54 | 0.46 | 0.59 | 0.26 | 10 |
| Gen-2 | 0.733 | 0.97 | 0.97 | 0.99 | **0.66** | 0.18 | 0.63 | 0.24 | 4 |
| LaVie | 0.746 | 0.91 | 0.97 | 0.96 | 0.54 | 0.49 | 0.61 | 0.26 | 3 |
| CogVideo | 0.673 | 0.92 | 0.95 | 0.96 | 0.38 | 0.42 | 0.41 | 0.07 | 4 |
| **Our Mora** | | | | | | | | | |
| Type I | 0.782 | 0.96 | 0.97 | 0.99 | 0.60 | 0.60 | 0.57 | 0.26 | 12 |
| Type II | 0.810 | 0.94 | 0.95 | 0.99 | 0.57 | 0.80 | 0.61 | 0.26 | 12 |
| Type III | 0.795 | 0.94 | 0.93 | 0.99 | 0.55 | 0.80 | 0.56 | 0.26 | 12 |
| Mora (SVD)$^{\mp}$ | 0.792 | 0.95 | 0.95 | 0.99 | 0.57 | **0.70** | 0.59 | 0.26 | 12 |
| Mora (Open-Sora-Plan)$^{\mp}$ | 0.767 | 0.94 | 0.95 | 0.99 | 0.61 | 0.43 | 0.68 | 0.26 | 12 |
| Mora (Open-Sora-Plan) | **0.800** | **0.98** | 0.97 | 0.99 | **0.66** | 0.50 | **0.70** | 0.31 | 12 |

0.70, matching Sora and surpassing all other comparative models, which clearly highlights the effectiveness of our multi-agent collaborative architecture. More impressively, finetuned Mora (OSP) models outperformed all baseline methods, including Sora, achieving state-of-the-art (SoTA) performance on multiple benchmarks. In the following experiment content, unless otherwise specified, Mora refers to Finetuned Mora (OSP). Mora demonstrates exceptional performance in maintaining overall consistency in video generation. Specifically, it excels in the Object Consistency task, achieving a leading score of 0.98 compared to other models. In terms of Background Consistency, Mora scores 0.97, comparable to the state-of-the-art performance of VideoCrafter1 (Chen et al., 2023a). This highlights Mora's strong capability to manage the overall coherence of generated videos. Mora achieved a Motion Smoothness score of 0.99, nearly matching Sora's perfect score of 1.0, demonstrating Mora's exceptional control over the smoothness of video sequences. It also outperformed in both Aesthetic Quality and Imaging Quality, with scores of 0.66 and 0.70, respectively. This reflects Mora's ability to not only ensure high imaging quality but also pursue a strong aesthetic dimension in its video generation. Our architecture and training method not only surpass Sora in performance but also achieve this with minimal computational resource requirements, showcasing exceptional optimization efficiency.

In Figure 2, we present examples of the generated videos. The visual fidelity of Mora's text-to-video generation is compelling, manifesting high-resolution imagery with acute attention to detail as articulated in the accompanying textual descriptions. Notably, the images exude a temporal consistency that speaks to Mora's nuanced understanding of narrative progression, an essential quality in video synthesis from textual prompts.

**Image-to-Video Generation.** Quantitative comparisons between Sora and Mora across different tasks are presented in Table 2. As shown in the table, Mora demonstrates comparable performance as Sora in the Motion Smoothness metric, achieving 0.98. In the VideoTI metric, both Mora and Sora are tied at 0.90. For the remaining metrics, Mora surpasses all other comparative methods, particularly achieving a perfect score of 1.0 in Dynamic Degree, demonstrating its exceptional capability in enhancing the sense of motion in videos. Additionally, Mora leads significantly in image quality with a score of 0.67, clearly indicating its superior performance in image rendering within video sequences. Also further demonstrates the usability and strong performance of Mora in the image-to-video task.

**Extend Generated Video.** The quantitative results from Table 2 reveal that Mora demonstrates similar performance as Sora in TCON and Temporal Style, with scores of 0.98 and 0.23 compared to Mora's 0.99 and 0.24. This indicates that Mora is nearly on par with Sora in maintaining stylistic continuity as well as sequence consistency and fidelity. In terms of Imaging Quality, Mora surpasses all other methods with scores of 0.45, demonstrating its excellent imaging capabilities. Despite Sora's very narrow lead, Mora effectively extends videos while maintaining high imaging quality, adhering

Table 2: Summary of model evaluations across various tasks based on the Sora technical report (OpenAI, 2024b).

| Model | Image-to-Video Generation | | | | Extend Videos | | |
|---|---|---|---|---|---|---|---|
| | VideoTI | Motion Smoothness | Dynamic Degree | Imaging Quality | TCON | Imaging Quality | Temporal Style |
| Sora | **0.90** | **0.99** | 0.75 | 0.63 | **0.99** | 0.43 | **0.24** |
| SVD | 0.88 | 0.97 | 0.75 | 0.58 | 0.94 | 0.37 | 0.22 |
| SVD-1.1 | 0.88 | 0.97 | 0.75 | 0.59 | 0.94 | 0.37 | 0.22 |
| Open-Sora-Plan | 0.89 | 0.98 | **1.00** | 0.65 | 0.96 | **0.45** | 0.22 |
| Mora (SVD) $^\mp$ | 0.88 | 0.97 | 0.75 | 0.60 | 0.94 | 0.39 | 0.22 |
| Mora (Open-Sora-Plan)$^\mp$ | 0.88 | 0.97 | **1.00** | 0.66 | 0.96 | **0.45** | 0.22 |
| Mora (Open-Sora-Plan) | **0.90** | 0.98 | **1.00** | **0.67** | 0.98 | **0.45** | 0.23 |

| Model | Video-to-Video Editing | | Connect Videos | | Simulate Digital Worlds | |
|---|---|---|---|---|---|---|
| | Imaging Quality | Temporal Style | Imaging Quality | Tmean | Imaging Quality | Appearance Style |
| Sora | **0.52** | **0.24** | **0.52** | **0.64** | **0.62** | **0.23** |
| Pika | 0.35 | 0.20 | - | - | 0.44 | 0.10 |
| Runway-tool | - | - | 0.33 | 0.22 | - | - |
| Mora (SVD) $^\mp$ | 0.38 | 0.23 | 0.42 | 0.45 | 0.52 | 0.23 |
| Mora (Open-Sora-Plan)$^\mp$ | 0.34 | 0.20 | 0.40 | 0.39 | 0.51 | 0.20 |
| Mora (Open-Sora-Plan) | **0.52** | **0.24** | 0.43 | 0.44 | 0.56 | **0.23** |

to temporal style, and ensuring strong stylistic continuity and sequence consistency, highlighting its effectiveness in the video extension domain.

**Video-to-Video Editing** Table 2 shows that Mora achieves respective scores of 0.52 and 0.24 in Imaging Quality and Temporal Style, respectively, matching Sora in both aspects. This indicates that Mora not only consistently achieves high-quality imaging in video editing but also effectively maintains stylistic consistency throughout. This also further highlights Mora's usability and strong adaptability in video-to-video editing tasks.

**Connect Videos.** Quantitative analysis in Table 2 shows that Sora outperforms Mora in both Imaging Quality and Tmean. Sora scores 0.52 in Imaging Quality and 0.64 in Temporal Coherence, while Mora's best scores are 0.43 and 0.45, respectively. This demonstrates Sora's superior fidelity in visual representation and consistency in visual narrative. These results highlight Sora's advantage in creating high-quality, temporally coherent video sequences, while also indicating areas where Mora could be further improved.

**Simulate Digital Worlds.** Table 2 shows that Sora leads in digital world simulation with a higher Imaging Quality score of 0.62, indicating better visual realism and fidelity compared to Mora's score of 0.52. However, both models achieve identical scores in Appearance Style at 0.23, indicating they equally adhere to the stylistic parameters of the digital world. This suggests that while there is a difference in the imaging quality, the stylistic translation of textual descriptions into visual aesthetics is accomplished with equivalent proficiency by both models.

More results, examples, ablation studies, and case studies are provided in Appendix B.

## 5 CONCLUSION

We introduce Mora, a pioneering generalist framework that synergies Standard Operating Procedures (SOPs) for video generation, tackling a broad spectrum of video-related tasks. Mora significantly advances the generation of videos from textual prompts, establishing new standards for adaptability, efficiency, and output quality in the domain. Our comprehensive evaluation indicates that Mora not only matches but also surpasses the capabilities of existing leading models in several respects. Nonetheless, it still faces a significant performance gap compared to OpenAI's Sora, whose closed-source nature presents substantial challenges for replication and further innovation in both academic and professional settings.

Looking forward, there are multiple promising avenues for further research. One direction includes enhancing the agents' natural language understanding capabilities to support more nuanced and context-aware video productions. Additionally, the issues of accessibility and high computational demands continue to impede broader adoption and innovation. Concurrently, fostering more open and collaborative research environments could spur advancements, allowing the community to build upon the foundational achievements of the Mora framework and other leading efforts.

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

# A  IMPLEMENTATION DETAILS

## A.1  HARDWARE DETAILS

All experiments are conducted on eight TESLA H100 GPUs, equipped with a substantial $8\times80$GB of VRAM. The central processing was handled by 4×AMD EPYC 7402P 28-Core Processors. Memory allocation was set at 320GB. The software environment was standardized on PyTorch version 2.0.2 and CUDA 12.2 for video generation and one TESLA A40, PyTorch version 1.10.2 and CUDA 11.6 for video evaluation.

## A.2  METRICS

**Basic Metrics.** ❶ Object Consistency, computed by the DINO (Caron et al., 2021) feature similarity across frames, assesses whether object appearance remains consistent throughout the entire video; ❷ Background Consistency, calculated by CLIP (Radford et al., 2021) feature similarity across frames; ❸ Motion Smoothness, utilizing motion priors in the video frame interpolation model AMT (Li et al., 2023) to evaluate the smoothness of generated motions; ❹ Aesthetic Score, obtained by using the LAION aesthetic predictor (lio) on each video frame to evaluate the artistic and beauty value perceived by humans, ❺ Dynamic Degree, computed by employing RAFT (Teed & Deng, 2020) to estimate the degree of dynamics in synthesized videos; ❻ Imaging Quality, calculated by MUSIQ (Ke et al., 2021) image quality predictor trained on SPAQ (Fang et al., 2020) dataset.

❶ Temporal Style, which is determined by utilizing ViCLIP (Wang et al., 2023b) to compute the similarity between video features and temporal style description features, thereby reflecting the consistency of the temporal style; ❷ Appearance Style, by calculating the feature similarity between synthesized frames and the input prompt using CLIP (Radford et al., 2021), to gauge the consistency of appearance style.

**Self-defined Metrics.** ❶ Video-Text Integration ($VideoTI$) employs LLaVA (Liu et al., 2024a) to transfer input image into textual descriptors $T_i$ and Video-Llama (Zhang et al., 2023b) to transfer videos generated by the model into textual $T_v$. The textual representation of the image is prepended with the original instructional text, forming an augmented textual input $T_{mix}$. Both the newly formed text and the video-generated text will be input to BERT (Devlin et al., 2018). The embeddings obtained are analyzed for semantic similarity through the computation of cosine similarity, providing a quantitative measurement of the model's adherence to the given instructions and image.

$$VideoTI = cosine(embed_{mix}, embed_v) \tag{1}$$

where $embed_{mix}$ represents the embedding for $T_{mix}$ and $embed_v$ for $T_v$.

❷ Temporal Consistency ($TCON$) assesses the integrity of extended video content. For each input-output video pair, we employ ViCLIP (Wang et al., 2023b) video encoder to extract their feature vectors. We then compute cosine similarity to get the score.

$$TCON = cosine(V_{input}, V_{output}) \tag{2}$$

❸ Temporal coherence $Tmean$ evaluates the temporal coherence by calculating the average correlation between intermediate generated videos and their neighboring input videos. Specifically, $TCON_{front}$ measures the correlation between the intermediate video and the preceding video in the time series, while $TCON_{beh}$ assesses the correlation with the subsequent video. The average of these scores provides an aggregate measure of temporal coherence across the video sequence.

$$Tmean = (TCON_{front} + TCON_{beh})/2 \tag{3}$$

## A.3  DETAILS OF SOPS

**Text-to-Video Generation.** This task harnesses a detailed textual prompt from the user as the foundation for video creation. The prompt must meticulously detail the envisioned scene. Utilizing this prompt, the Text-to-Image agent utilizes the text, distilling themes and visual details to craft an initial frame. Building upon this foundation, the Image-to-Video component methodically generates a sequence of images. This sequence dynamically evolves to embody the prompt's described actions

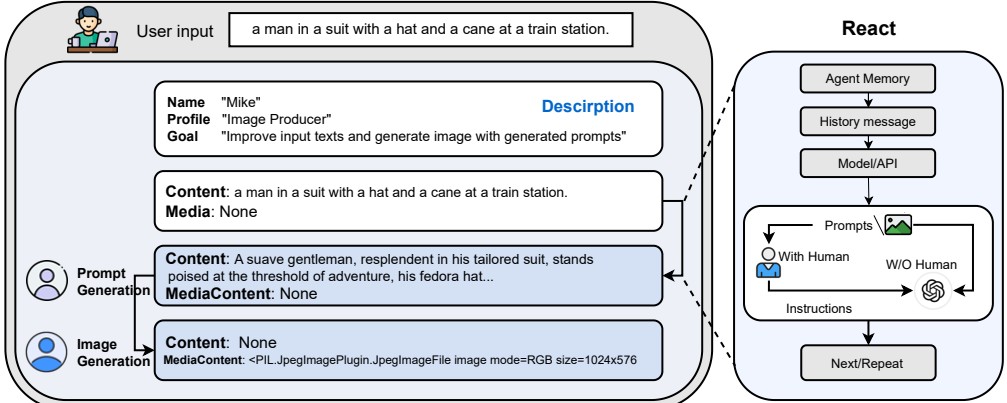

Figure 5: An example of image generation process in Mora. Left: Agent uses the structured message to communicate, Right: After the prompt or image is generated, a human user can check the quality of the generated content.

or scenes, and each video is derived from the last frame from the previous video, thereby achieving a seamless transition throughout the video.

**Image-to-Video Generation.** This task mirrors the operational pipeline of Task 1, with a key distinction. Unlike Task 1 with only texts as inputs, Task 2 integrates both a textual prompt and an initial image into the Text-to-Image agent's input. This dual-input approach enriches the content generation process, enabling a more nuanced interpretation of the user's vision.

**Extend Generated Videos.** This task focuses on extending the narrative of an existing video sequence. By taking the last frame of an input video as the starting point, the video generation agent crafts a series of new, coherent frames that continue the story. This approach allows for the seamless expansion of video content, creating longer narratives that maintain the consistency and flow of the original sequence without disrupting its integrity.

**Video-to-Video Editing.** This task introduces a sophisticated editing capability, leveraging both the Image-to-Image and Image-to-Video agents. The process begins with the Image-to-Image agent, which takes the first frame of an input video and applies edits based on the user's prompt, achieving the desired modifications. This edited frame then serves as the initial image for the Image-to-Video agent, which generates a new video sequence that reflects the requested obvious or subtle changes, offering a powerful tool for dynamic video editing.

**Connect Videos.** The Image-to-Video agent leverages the final frame of the first input video and the initial frame of the second input video to create a seamless transition, producing a new video that smoothly connects the two original videos.

**Simulating Digital Worlds.** This task specializes in the whole style changing for video sequences set in digitally styled worlds. By appending the phrase "In digital world style" to the edit prompt, the user instructs the Image-to-Video agent to craft a sequence that embodies the aesthetics and dynamics of a digital realm or utilize the Image-to-Image agent to transfer the real image to digital style. This task pushes the boundaries of video generation, enabling the creation of immersive digital environments that offer a unique visual experience.

A.4  ITERATIVE PROGRAMMING WITH HUMAN IN THE LOOP

Human oversight and iterative refinement are essential in content generation tasks, improving the quality and precision of the final outputs. Integrating human collaboration within video generation frameworks is pivotal for ensuring that the resulting videos conform to the expected standards.

As illustrated in Figure 1 and 5, Mora is engineered to generate videos based on input prompts while executing SOPs under continuous human supervision. This supervisory role enables users to rigorously monitor the process and verify that the outputs align with their expectations. Following each stage, users have the discretion to either advance to the subsequent phase or request a repetition

of the previous one, with a maximum of three iterations allowed per stage. This iterative mechanism persists until the output is deemed satisfactory, ensuring high fidelity to user requirements.

## A.5 FUNCTION OF AGENTS

**Prompt Selection and Generation Agent.** Prior to the commencement of the initial image generation, textual prompts undergo a rigorous processing and optimization phase. This critical agent can employ large language models like GPT-4, GPT-4o, Llama, Llama3 (Achiam et al., 2023; GPT; Touvron et al., 2023). It is designed to meticulously analyze the text, extracting pivotal information and actions delineated within, thereby significantly enhancing the relevance and quality of the resultant images. This step ensures that the textual descriptions are thoroughly prepared for an efficient and effective translation into visual representations.

**Text-to-Image Generation Agent.** The text-to-image model (Rombach et al., 2022c; Podell et al., 2023) stands at the forefront of translating these enriched textual descriptions into high-quality initial images. Its core functionality revolves around a deep understanding and visualization of complex textual inputs, enabling it to craft detailed and accurate visual counterparts to the provided textual descriptions.

**Image-to-Image Generation Agent.** This agent (Brooks et al., 2023) works to modify a given source image in response to specific textual instructions. The core of its functionality lies in its ability to interpret detailed textual prompts with high accuracy and subsequently apply these insights to manipulate the source image accordingly. This involves a detailed recognition of the text's intent, translating these instructions into visual modifications that can range from subtle alterations to transformative changes. The agent leverages a pre-trained model to bridge the gap between textual description and visual representation, enabling seamless integration of new elements, adjustment of visual styles, or alteration of compositional aspects within the image.

**Image-to-Video Generation Agent.** Following the creation of the initial image, the Video Generation Model (Blattmann et al., 2023) is responsible for transitioning the static frame into a vibrant video sequence. This component delves into the analysis of both the content and style of the initial image, serving as the foundation for generating subsequent frames. These frames are meticulously crafted to ensure a seamless narrative flow, resulting in a coherent video that upholds temporal stability and visual consistency throughout. This process highlights the model's capability to not only understand and replicate the initial image but also to anticipate and execute logical progressions in the scene.

**Video Connection Agent.** Utilizing the Video-to-Video Agent, we create seamless transition videos based on two input videos provided by users. This advanced agent selectively leverages key frames from each input video to ensure a smooth and visually consistent transition between them. It is designed with the capability to accurately identify the common elements and styles across the two videos, thus ensuring a coherent and visually appealing output. This method not only improves the seamless flow between different video segments but also retains the distinct styles of each segment.

## A.6 MLLMS TO JUDGE VIDEO QUALITY

We employ two multimodal large language models (MLLMs) to evaluate four videos, using random prompts selected from Table 3. The MLLMs process the inputs and generate rankings based on the specified evaluation criteria. From the output, we focus on analyzing the video ranked as top-1 by each MLLM. If both MLLMs agree on the top-1 video, it is directly included in the training dataset. In cases where the MLLMs disagree, human reviewers intervene to further assess the videos and finalize the selection. This hybrid approach ensures the inclusion of high-quality, relevant videos, while balancing automation with human judgment. By integrating MLLMs with human oversight, we maintain robustness and precision in the data selection process, which ultimately enhances the video generation model's performance.

## A.7 DETAILS OF TRAINING PIPELINE

The training pipeline is illustrated in two key phases, as shown in Figure 6, constructing the training dataset and self-modulated fine-tuning.

Table 3: Evaluation Prompts for MLLMs to Judge Video Quality

| Prompt No. | Evaluation Criteria |
|---|---|
| 1 | Evaluate the visual clarity and resolution, ranking videos based on image sharpness, smoothness of transitions, and noise levels. |
| 2 | Assess object consistency and scene stability across frames, ranking videos on object motion and interactions. |
| 3 | Examine the temporal coherence, identifying the best frame-to-frame continuity. |
| 4 | Evaluate the narrative coherence or logical progression, ranking based on storytelling consistency. |
| 5 | Assess color grading and lighting consistency, determining the best video based on smooth lighting transitions and uniform color. |
| 6 | Evaluate the realism of objects, background textures, and scene complexity, ranking videos from most realistic to least. |
| 7 | Analyze content relevance to the task, ranking videos based on theme alignment and task appropriateness. |
| 8 | Compare the aesthetic quality, focusing on artistic composition, balance, and overall visual appeal. |
| 9 | Evaluate noise and artifact levels, identifying the video with the cleanest and smoothest output. |
| 10 | Examine frame rate consistency and smoothness of motion, ranking videos based on natural motion without lag or stuttering. |

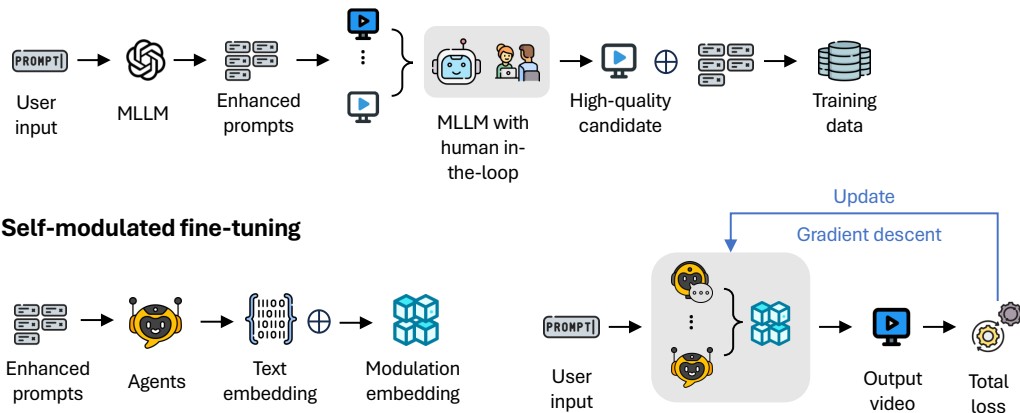

Figure 6: Illustration of the process of constructing training dataset and the design of our self-modulated fine-tuning.

**Constructing the Training Dataset.** The process starts with user input prompts that are enhanced by a multimodal large language model (MLLM). These enhanced prompts are evaluated using a human-in-the-loop strategy, which ensures high-quality video candidates. The selected videos, paired with their prompts, form the final training data for fine-tuning the model.

**Self-Modulated Fine-Tuning.** During fine-tuning, enhanced prompts are processed by multiple agents. Each agent generates text embeddings, which are combined with a modulation embedding to dynamically adjust the contribution of each agent. The agents work together in a coordinated manner to generate video outputs. The total loss is calculated based on the output video and the target, and gradient descent is used to update both the model parameters and the modulation embeddings, and then reconstruct the training datasets. This ensures that the agents adapt their outputs based on the preceding agent's state, resulting in improved video generation.

A.8 DETAILS OF AGENTS

**Prompt Selection and Generation**. Currently, GPT-4o (GPT) stands as the most advanced generative model available. By harnessing the capabilities of GPT-4o, we are able to generate and meticulously select high-quality prompts. These prompts are detailed and rich in information, facilitating the Text-to-Image generation process by providing the agent with comprehensive guidance.

**Text-to-Image Generation**. We utilize the pre-trained large text-to-image model to generate a high-quality and representative first image.

For Mora (SVD), the initial implementation utilizes Stable Diffusion XL (SDXL) (Podell et al., 2023). It introduces a significant evolution in the architecture and methodology of latent diffusion models (Rombach et al., 2022c; Meng et al., 2022) for text-to-image synthesis, setting a new benchmark in the field. At the core of its architecture is an enlarged UNet backbone (Ronneberger et al., 2015) that is three times larger than those used in previous versions of Stable Diffusion 2 (Rombach et al., 2022c). This expansion is principally achieved through an increased number of attention blocks and a broader cross-attention context, facilitated by integrating a dual text encoder system. The first encoder is based on OpenCLIP (Ilharco et al., 2021) ViT-bigG (Cherti et al., 2023; Radford et al., 2021; Schuhmann et al., 2022), while the second utilizes CLIP ViT-L, allowing for a richer, more nuanced interpretation of textual inputs by concatenating the outputs of these encoders. This architectural innovation is complemented by the introduction of several novel conditioning schemes that do not require external supervision, enhancing the model's flexibility and capability to generate images across multiple aspect ratios. Moreover, SDXL features a refinement model that employs a post-hoc image-to-image transformation to elevate the visual quality of the generated images. This refinement process utilizes a noising-denoising technique, further polishing the output images without compromising the efficiency or speed of the generation process.

For Mora (Open-Sora-Plan), the Stable Diffusion 3 (SD3) (Esser et al., 2024) text-to-image model employed to generate high-quality images from textual prompts. SD3 utilizes a rectified flow transformer architecture, offering significant advancements over traditional diffusion models by connecting data and noise in a linear path rather than the curved trajectories commonly used in previous approaches. This architectural choice facilitates a more efficient and accurate sampling process, enhancing the generation of high-resolution images across diverse styles. Additionally, SD3 introduces a novel noise re-weighting technique that biases sampling toward perceptually relevant scales, thereby improving the clarity and aesthetic quality of the generated images. The model also supports multiple aspect ratios and incorporates a refinement process that further enhances generated images through a noising-denoising technique, ensuring high visual fidelity while maintaining computational efficiency.

**Image-to-Image Generation**. Our initial Mora (SVD) framework utilizes InstructPix2Pix as Image-to-Image generation agent. InstructPix2Pix (Brooks et al., 2023) are intricately designed to enable effective image editing from natural language instructions. At its core, the system integrates the expansive knowledge of two pre-trained models: GPT-3 (Brown et al., 2020) for generating editing instructions and edited captions from textual descriptions, and Stable Diffusion (Rombach et al., 2022b) for transforming these text-based inputs into visual outputs. This ingenious approach begins with fine-tuning GPT-3 on a curated dataset of image captions and corresponding edit instructions, resulting in a model that can creatively suggest plausible edits and generate modified captions. Following this, the Stable Diffusion model, augmented with the Prompt-to-Prompt technique, generates pairs of images (before and after the edit) based on the captions produced by GPT-3. The conditional diffusion model at the heart of InstructPix2Pix is then trained on this generated dataset. InstructPix2Pix directly utilizes the text instructions and input image to perform the edit in a single forward pass. This efficiency is further enhanced by employing classifier-free guidance for both the image and instruction conditionings, allowing the model to balance fidelity to the original image with adherence to the editing instructions.

For Mora (Open-Sora-Plan), the image-to-image generation agent, based on SD3 (Esser et al., 2024), excels in applying detailed text-guided transformations to images with precision and flexibility. SD3's rectified flow transformer architecture ensures efficient and accurate image modifications, while the noise re-weighting process enhances the visual quality of the output, ensuring that edits are seamless and coherent.

**Image-to-Video Generation**. In the Text-to-Video generation agent, video generation agents play an important role in ensuring video quality and consistency.

Our initial Mora (SVD) implementation utilizes the state-of-the-art video generation model Stable Video Diffusion to generate video. The Stable Video Diffusion (SVD) (Blattmann et al., 2023) architecture introduces a cutting-edge approach to generating high-resolution videos by leveraging the strengths of LDMs Stable Diffusion v2.1 (Rombach et al., 2022b), originally developed for image synthesis, and extending their capabilities to handle the temporal complexities inherent in video content. At its core, the SVD model follows a three-stage training regime that begins with text-to-image pertaining, where the model learns robust visual representations from a diverse set of images. This foundation allows the model to understand and generate complex visual patterns and textures. In the second stage, video pretraining, the model is exposed to large amounts of video data, enabling it to learn temporal dynamics and motion patterns by incorporating temporal convolution and attention layers alongside its spatial counterparts. This training is conducted on a systematically curated dataset, ensuring the model learns from high-quality and relevant video content. The final stage, high-quality video finetuning, focuses on refining the model's ability to generate videos with increased resolution and fidelity, using a smaller but higher-quality dataset. This hierarchical training strategy, complemented by a novel data curation process, allows SVD to excel in producing state-of-the-art text-to-video and image-to-video synthesis with remarkable detail, realism, and coherence over time.

The Mora (Open-Sora-Plan) video generation agent leverages a 3D full attention mechanism, which processes spatial and temporal features simultaneously, providing a unified understanding of motion and appearance across frames. This mechanism ensures smoother transitions and higher coherence in movements across consecutive frames, improving the temporal consistency of the generated video. In the second phase, temporal convolution layers are integrated to further refine the model's capacity to comprehend and generate realistic motion dynamics. By capturing temporal patterns over time, the model produces videos with continuous and natural movements, thereby enhancing the realism of complex visual sequences. The third phase involves utilizing the CausalVideoVAE (Chen et al., 2024b), which enhances the visual representation and detail of each frame while minimizing the occurrence of artifacts commonly associated with video synthesis. The CausalVideoVAE maintains high-quality visual outputs throughout the entire video sequence by refining and polishing the generated content through multiple iterations. This comprehensive training regime ensures that Mora (Open Sora Plan) excels in both image-to-video and text-to-video tasks, delivering exceptional levels of detail, realism, and temporal coherence. These architectural innovations enable superior handling of complex motion patterns and high-resolution video outputs, resulting in visually consistent and temporally stable videos.

**Connect Videos.** For the video connection task, we utilize SEINE (Chen et al., 2023c) to connect videos. SEINE is constructed upon a pre-trained diffusion-based T2V model, LaVie (Wang et al., 2023a) agent. SEINE centered around a random-mask video diffusion model that generates transitions based on textual descriptions. By integrating images of different scenes with text-based control, SEINE produces transition videos that maintain coherence and visual quality. Additionally, the model can be extended for tasks such as image-to-video animation and autoregressive video prediction.

In Mora (Open-Sora-Plan), the video connection task utilizes a specialized architecture to ensure seamless and visually coherent transitions between videos. This is achieved by combining diffusion-based models with temporal convolution techniques, facilitating smooth transitions in style, content, and motion dynamics. The video connection agent employs CausalVideoVAE, optimized for temporal and spatial consistency, enhancing the connection of two video segments by identifying and preserving common visual elements across frames. A 3D full attention architecture is central to this process, enabling the model to simultaneously capture spatial and temporal features, thereby maintaining coherence in object motion and background continuity. Additionally, the architecture incorporates random-mask video diffusion, which maintains high-resolution transitions by infilling missing information based on text-based control inputs or video context. This approach ensures that the connected videos preserve visual quality and exhibit coherent motion patterns, resulting in high-quality, temporally stable transitions.

## A.9 DETAILS SETTINGS IN TRAINING

In our training setup, we use the `AdamW` optimizer, which is known for handling weight decay effectively, with an initial learning rate of `1e-5`. The learning rate remains constant throughout training, controlled by the `constant` scheduler, and no warmup steps are used (`lr_warmup_steps=0`). To handle memory constraints, we accumulate gradients over 1 step, which allows us to use a batch size of `4` while maintaining stable optimization. We enable gradient checkpointing to further save memory by reducing the number of intermediate activations stored during backpropagation, although this comes at the cost of slower backward passes. Mixed precision training with `bf16` is employed to enhance computational speed and lower memory usage, which is crucial when training with a batch size of `4`. The model is trained for a maximum of `12` steps, with checkpoints being saved every `4` steps to allow for model recovery or evaluation during training. We use an `SNR Gamma` value of `5.0` to balance the noise scale, which is especially useful for diffusion-based models, and apply Exponential Moving Average (EMA) from step `0` with a decay rate of `0.999` to ensure model stability throughout training. These settings provide a balance between computational efficiency and model performance, ensuring that we can handle large video data with high memory demands while optimizing effectively across training iterations.

Since the OpenSora and SD3 environments are mutually exclusive (due to certain package incompatibilities), we use inter-process communication (IPC) to perform joint training.

Listing 1: SD3 Process (Text-to-Image) Pseudocode

```
# Initialize ZeroMQ context and create a REQ socket
Initialize context
Create REQ socket
Connect to Open-Sora-Plan environment

# Main loop
while training is not complete:
    # Generate image from Stable-Diffusion3 with prompt
    image = Stable_Diffusion3(prompt)

    # Send image data to Open-Sora-Plan
    socket.send(image)

    # Receive video loss from Open-Sora-Plan
    video_loss = socket.recv()

    # Backpropagation
    optimizer.zero_grad()
    video_loss.backward()
    optimizer.step()
```

Listing 2: Open-Sora-Plan Process (Image-to-Video) Pseudocode

```
# Initialize ZeroMQ context and create a REP socket
Initialize context
Create REP socket
Bind to specific port

# Main loop
while True:
    # Receive image data from SD3
    image_data = socket.recv()

    # Process the image data
    image = process_image(image_data)

    # Generate video from image using Open-Sora-Plan
    generated_video = Open_Sora_Plan(image, prompts)

    # Compute video loss
    video_loss = compute_loss(generated_video, ground_truth_video)
```

```
    # Send video loss back to SD3
    socket.send(video_loss)
```

## A.10 MODULATION VISUALIZATION

To evaluate the effectiveness of self-modulated fine-tuning on the Text-to-Video generation task, we visualize the modulation features in Figure 7. The heatmaps represent the activation distributions of modulation features across different model configurations and setups. The top row of the figure corresponds to results without self-modulated fine-tuning, while the bottom row illustrates the outcomes with self-modulated fine-tuning applied. Each column depicts the feature maps for modulations of various agents: A2: SD3, A3: SD3, and A4: Open-Sora-Plan. The heatmap intensity (color bar on the left) ranges from 0 (purple) to 1 (yellow), representing the activation magnitudes. The sharper contrasts and concentrated patterns in the fine-tuned models further validate the utility of the proposed method. Without self-modulated fine-tuning, the

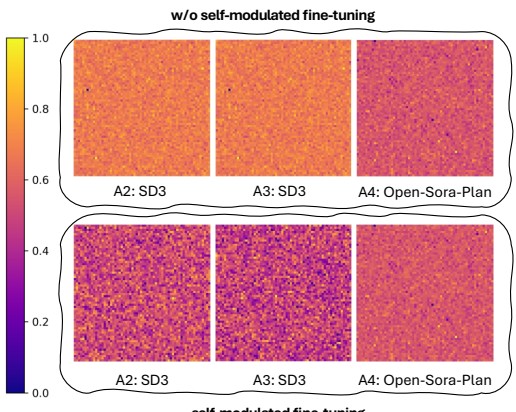

Figure 7: Visualization of Modulation Feature in self-modulated fine-tuning on Text-to-Video task

modulation values for A2, A3, and A4 were 6.667, 6.667, and 9.630, respectively. After training with self-modulated fine-tuning, these values were updated to 4.330, 1.789, and 12.335. This demonstrates that the model progressively allocates greater focus to A4 (the Image-to-Video Agent) during training, suggesting its increased importance in the task. Specifically, the significant increase in A4's modulation value highlights its critical role in enhancing the Text-to-Video generation quality. This finding further supports self-modulated fine-tuning promoting effective collaboration among agents, enabling the framework to balance and optimize contributions from each agent adaptively.

## A.11 TRAINING SAMPLES STUDY

In this subsection, we analyze the impact of training sample size on the performance of our system to address concerns regarding the sufficiency of using only 96 samples during training. The experiments presented in Table A illustrate the results of varying the sample size from 16 to 144 while measuring the average performance across six tasks.

The results as shown in Table 4 illustrate that while increasing the number of samples leads to performance improvements initially, the gains plateau as the sample size approaches 96. Specifically, the average performance metric (**Avg.**) increases from 0.6059 with 16 samples to a peak of 0.6482 at 96 samples, with only negligible fluctuations thereafter. This plateau suggests that our training strategy effectively maximizes the utility of the provided samples, leveraging their representational power to enhance inter-agent collaboration and alignment.

The diminishing returns beyond 96 samples can be attributed to our self-modulated fine-tuning algorithm, which optimizes agent interactions using a data-free training strategy. By focusing on refining the agents' pre-trained capabilities and aligning their outputs to a shared understanding, the algorithm ensures efficient use of limited training data. This efficiency is further reinforced by the human-in-the-loop mechanism, which guides the optimization process, making additional data redundant after a certain threshold.

## B MORE RESULTS AND EXAMPLES

### B.1 MORE RESULTS DETAILS

Table 4: Performance of Mora with training setup of different sample sizes.

| Samples | Task 1 (Avg.) | Task 2 (Avg.) | Task 3 (TCON) | Task 4 (Avg.) | Task 5 (Avg.) | Task 6 (IQ) | Avg. |
|---|---|---|---|---|---|---|---|
| 16 | 0.767 | 0.8775 | 0.9621 | 0.27 | 0.401 | 0.353 | 0.6059 |
| 32 | 0.788 | 0.8806 | 0.9721 | 0.33 | 0.423 | 0.380 | 0.6298 |
| 48 | 0.793 | 0.8830 | 0.9766 | 0.36 | 0.437 | 0.394 | 0.6404 |
| 64 | 0.797 | 0.8850 | 0.9802 | 0.37 | 0.441 | 0.396 | 0.6447 |
| 80 | 0.799 | 0.8868 | 0.9820 | 0.37 | 0.442 | 0.399 | 0.6468 |
| 96 | 0.800 | 0.8874 | 0.9830 | 0.38 | 0.442 | 0.399 | **0.6482** |
| 112 | 0.800 | 0.8875 | 0.9835 | 0.38 | 0.440 | 0.398 | 0.6480 |
| 128 | 0.799 | 0.8875 | 0.9837 | 0.37 | 0.441 | 0.398 | 0.6469 |
| 144 | 0.800 | 0.8876 | 0.9836 | 0.37 | 0.442 | 0.398 | 0.6480 |

**Text-to-Video Generation.** The quantitative results are detailed in Table 1 and Figure 8. Mora showcases commendable performance across all metrics, making it highly comparable to the top-performing model, Sora, and surpassing the capabilities of other competitors. Specifically, Mora achieved a Video Quality score of 0.792, which closely follows Sora's leading score of 0.797 and surpasses the current best open-source model like VideoCrafter1. In terms of Object Consistency, Mora scored 0.95, equaling Sora and demonstrating superior consistency in maintaining object identities throughout the videos. For Background Consistency and Motion Smoothness, Mora achieved scores of 0.95 and 0.99, respectively, indicating high fidelity in background stability and fluidity of motion within generated videos. Although Sora achieved 0.96 slightly outperforms Mora in Background Consistency, the margin is minimal. The Aesthetic Quality metric, which assesses

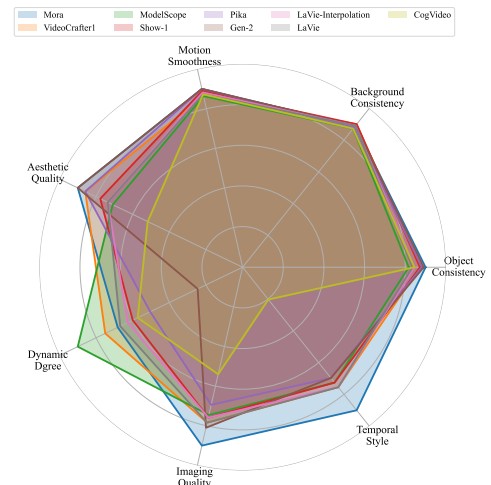

Figure 8: Comparative analysis of text-to-video generation performance between Mora and various other models.

the overall visual appeal of the videos, saw Mora scoring 0.57. This score, while not the highest, reflects a competitive stance against other models, with Sora scoring slightly higher at 0.60. Nevertheless, Mora's performance in Dynamic Degree and Imaging Quality, with scores of 0.70 and 0.59, showcases its strength in generating dynamic, visually compelling content that surpasses all other models. As for Temporal Style, Mora scored 0.26, indicating its robust capability in addressing the temporal aspects of video generation. Although this performance signifies a commendable proficiency, it also highlights a considerable gap between our model and Sora, the leader in this category with a score of 0.35.

The results in Table 5 compare various models on video generation performance using the Vbench dataset. Mora (Open-Sora-Plan) achieves the highest overall video quality (0.848) and excels in both object and background consistency (0.98), while also leading in aesthetic quality (0.70) and imaging quality (0.72). Emu3 and Gen-3-Alpha perform similarly in video quality (0.841), with Emu3 showing superior motion smoothness (0.99) and dynamic degree (0.79). Despite a slightly lower video quality, LaVie-2 outperforms other models in temporal style and maintains strong consistency metrics across the board.

In Figure 2, the visual fidelity of Mora's text-to-video generation is compelling, manifesting high-resolution imagery with acute attention to detail as articulated in the accompanying textual descriptions. The vivid portrayal of scenes, from the liftoff of a rocket to the dynamic coral ecosystem and the urban skateboarding vignette, underscores the system's adeptness in capturing and translating the essence of the described activities and environments into visually coherent sequences. Notably, the images exude a temporal consistency that speaks to Mora's nuanced understanding of narrative progression, an essential quality in video synthesis from textual prompts.

Mora can deal with various tasks, including video editing, video extension, and stimulate digital world. We provide demo of Mora in Figure 9.

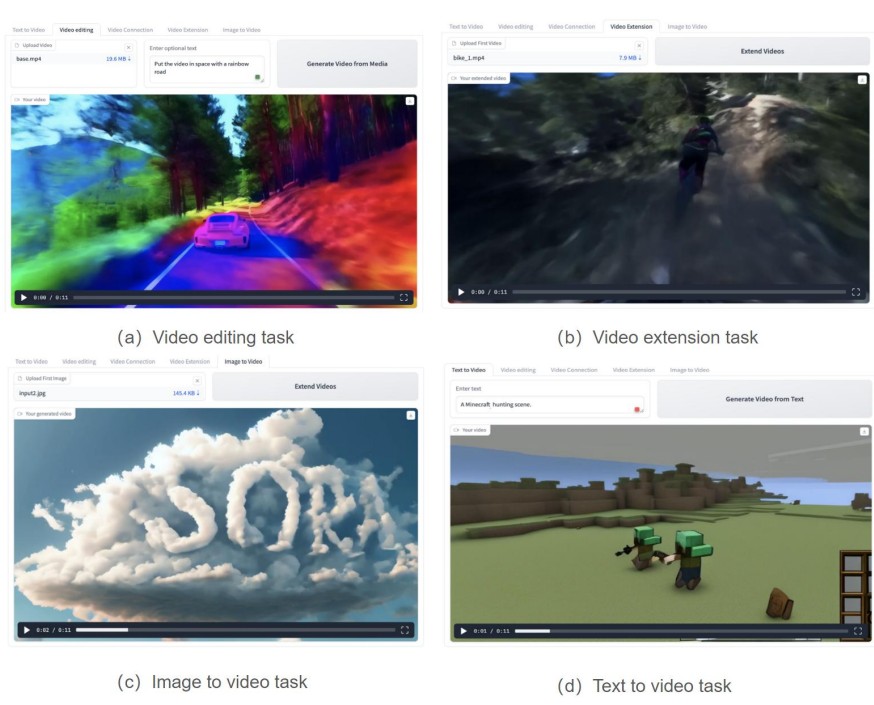

(a) Video editing task          (b) Video extension task

(c) Image to video task          (d) Text to video task

Figure 9: Demo of Mora on various tasks.

Table 5: Comparative analysis of text-to-video generation performance for models with Vbench data.

| Model | Video Quality | Object Consistency | Background Consistency | Motion Smoothness | Aesthetic Quality | Dynamic Degree | Imaging Quality | Temporal Style | Video Length(s) |
|---|---|---|---|---|---|---|---|---|---|
| Emu3 | 0.841 | 0.95 | 0.98 | **0.99** | 0.60 | **0.79** | 0.63 | 0.24 | 5 |
| Open-Sora-Plan_V1.2 | 0.814 | 0.97 | **0.98** | **0.99** | 0.59 | 0.42 | 0.57 | 0.25 | 8 |
| Gen-3-Alpha | 0.841 | 0.97 | 0.97 | **0.99** | 0.63 | 0.60 | 0.67 | 0.25 | 10 |
| LaVie-2 | 0.832 | **0.98** | **0.98** | 0.98 | 0.68 | 0.31 | 0.70 | 0.25 | 3 |
| Mora (Open-Sora-Plan) | **0.848** | **0.98** | **0.98** | **0.99** | **0.70** | 0.72 | **0.72** | 0.25 | 12 |

**Text-conditional Image-to-Video Generation.** In Figure 15, a qualitative comparison between the video outputs from Sora and Mora reveals that both models adeptly incorporate elements from the input prompt and image. The monster illustration and the cloud spelling "SORA" are well-preserved and dynamically translated into video by both models. Despite quantitative differences, the qualitative results of Mora nearly rival those of Sora, with both models are able to animate the static imagery and narrative elements of the text descriptions into coherent video. This qualitative observation attests to Mora's capacity to generate videos that closely parallel Sora's output, achieving a high level of performance in rendering text-conditional imagery into video format while maintaining the thematic and aesthetic essence of the original inputs.

**Extend Generated Videos.** From a qualitative standpoint, Figure 11 illustrates the competencies of Mora in extending video sequences. Both Sora and Mora adeptly maintain the narrative flow and visual continuity from the original to the extended video. Despite the slight numerical differences highlighted in the quantitative analysis, the qualitative outputs suggest that Mora's extended videos preserve the essence of the original content with high fidelity. The preservation of dynamic elements such as the rider's motion and the surrounding environment's blur effect in the Mora generated sequences showcases its capacity to produce extended videos that are not only coherent but also retain the original's motion and energy characteristics. This visual assessment underscores Mora's proficiency in generating extended video content that closely mirrors the original, maintaining the narrative context and visual integrity, thus providing near parity with Sora's performance.

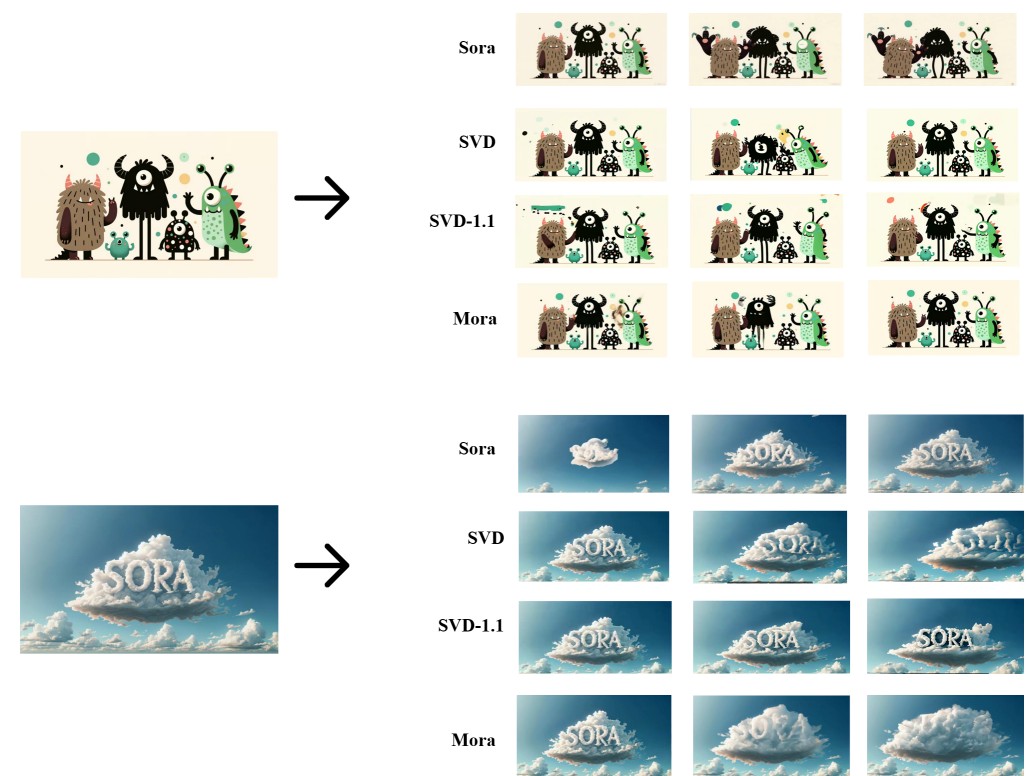

Figure 10: Samples for text-conditional image-to-video generation of Mora and Sora. Prompt for the first line image is: Monster Illustration in flat design style of a diverse family of monsters. The group includes a furry brown monster, a sleek black monster with antennas, a spotted green monster, and a tiny polka-dotted monster, all interacting in a playful environment. The second image's prompt is: An image of a realistic cloud that spells "SORA".

**Video-to-Video Editing.** Upon qualitative evaluation, Figure 12 presents samples from video-to-video editing tasks, wherein both Sora and Mora were instructed to modify the setting to the 1920s style while maintaining the car's red color. Visually, Sora's output exhibits a transformation that convincingly alters the modern-day setting into one reminiscent of the 1920s, while carefully preserving the red color of the car. Mora's transformation, while achieving the task instruction, reveals differences in the execution of the environmental modification, with the sampled frame from generated video suggesting a potential for further enhancement to achieve the visual authenticity displayed by Sora. Nevertheless, Mora 's adherence to the specified red color of the car underline its ability to follow detailed instructions and enact considerable changes in the video content. This capability, although not as refined as Sora's, demonstrates Mora's potential for significant video editing tasks.

**Connect Videos.** Qualitative analysis based on Figure 13 suggest that, in comparison to Sora's proficiency in synthesizing intermediate video segments that successfully incorporate background elements from preceding footage and distinct objects from subsequent frames within a single frame, the Mora model demonstrates a blurred background in the intermediate videos, which results in indistinguishable object recognition. Accordingly, this emphasizes the potential for advancing the fidelity of images within the generated intermediate videos as well as enhancing the consistency with the entire video sequence. This would contribute to refining the video connecting process and improving the integration quality of Mora's model outputs.

**Simulate Digital Worlds.** Upon qualitative evaluation, Figure 14 presents samples from Simulate digital worlds tasks, wherein both Sora and Mora were instructed to generated video of "Minecraft" scenes. In the top row of frames generated by Sora, we see that the videos maintain high fidelity to the textures and elements typical of digital world aesthetics, characterized by crisp edges, vibrant

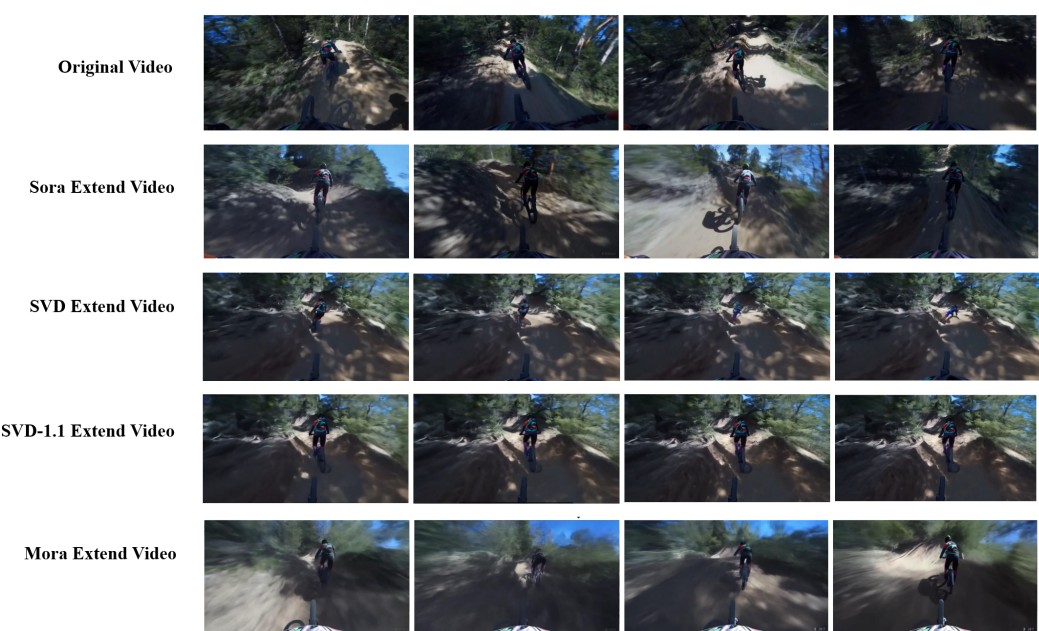

Figure 11: Samples for Extend generated video of Mora and Sora.

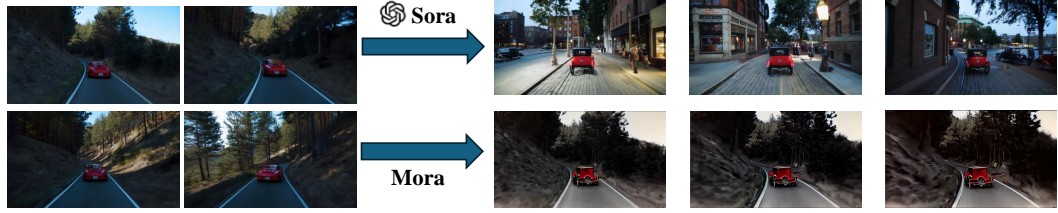

Instruction: Change the setting to the 1920s with an old school car. make sure to keep the red color

Figure 12: Samples for Video-to-video editing

colors, and clear object definition. The pig and the surrounding environment appear to adhere closely to the style one would expect from a high-resolution game or a digital simulation. These are crucial aspects of performance for Sora, indicating a high-quality synthesis that aligns well with user input while preserving visual consistency and digital authenticity. The bottom row of frames generated by Mora suggests a step towards achieving the digital simulation quality of Sora but with notable differences. Although Mora seems to emulate the digital world's theme effectively, there is a visible gap in visual fidelity. The images generated by Mora exhibit a slightly muted color palette, less distinct object edges, and a seemingly lower resolution compared to Sora's output. This suggests that Mora is still in a developmental phase, with its generative capabilities requiring further refinement to reach the performance level of Sora.

## B.2    ABLATION STUDY

The results in Table 6 clearly demonstrate the effectiveness of each component in our Mora model. The full version of Mora (Open-Sora-Plan) outperforms all ablated variants across most metrics, showcasing the importance of each design choice. First, Mora achieves the highest video quality score of 0.800, along with the best object consistency (0.98), background consistency (0.97), motion smoothness (0.99), and imaging quality (0.70). This underscores the significance of incorporating all components, as removing any one of them leads to a noticeable drop in performance. For instance, removing the human-in-the-loop module (Mora w/o Human-in-the-loop) reduces video quality to

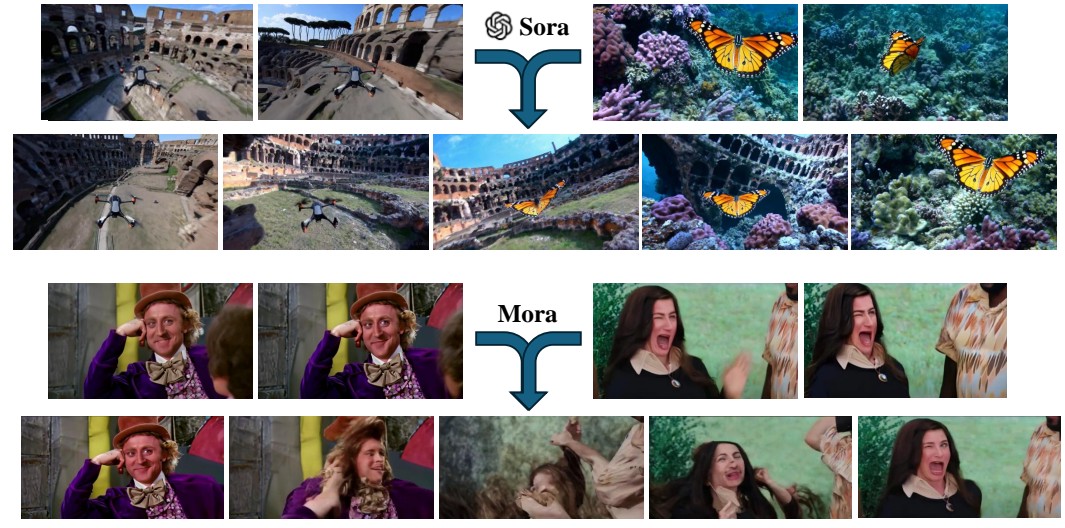

Figure 13: Samples for Video Connetion

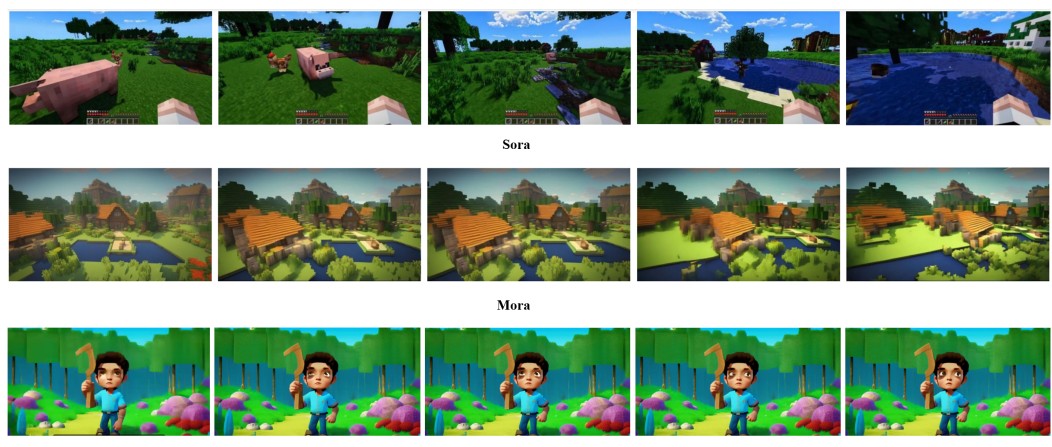

Figure 14: Samples for Simulate digital worlds

0.785 and slightly decreases motion smoothness (0.95), indicating that human guidance is key to maintaining higher video generation quality and consistency. Similarly, removing self-modulation (Mora w/o Self-modulated) results in a further decline in video quality to 0.776, while slightly improving the dynamic degree (0.51). However, this increase in dynamic degree comes at the cost of consistency across other metrics, showing that self-modulation balances aesthetic quality and consistency. The Random Initial modulation (Mora with RI-modulated) also demonstrates a strong

Table 6: Ablation study on different variants of Mora model for text-to-video generation performance. The Random Initial modulation (RI-modulated) embeddings represent $\{\mathbf{z}_i \in \mathbb{R}^{1 \times \text{text\_encoder}_i\_\text{feature}}\}_{i=1}^n$.

| Model | Video Quality | Object Consistency | Background Consistency | Motion Smoothness | Aesthetic Quality | Dynamic Degree | Imaging Quality | Temporal Style |
|---|---|---|---|---|---|---|---|---|
| Mora (Open-Sora-Plan) | **0.800** | **0.98** | **0.97** | **0.99** | **0.66** | 0.50 | **0.70** | **0.31** |
| Mora (Open-Sora-Plan)∓ | 0.767 | 0.94 | 0.95 | 0.99 | 0.61 | 0.43 | 0.68 | 0.26 |
| Mora w/o Human-in-the-loop | 0.785 | 0.98 | 0.96 | 0.95 | 0.63 | 0.50 | 0.69 | 0.31 |
| Mora w/o Self-modulated | 0.776 | 0.96 | 0.95 | 0.95 | 0.62 | **0.51** | 0.67 | 0.27 |
| Mora with RI-modulated | 0.797 | 0.98 | 0.96 | 0.99 | 0.66 | 0.50 | 0.69 | 0.29 |

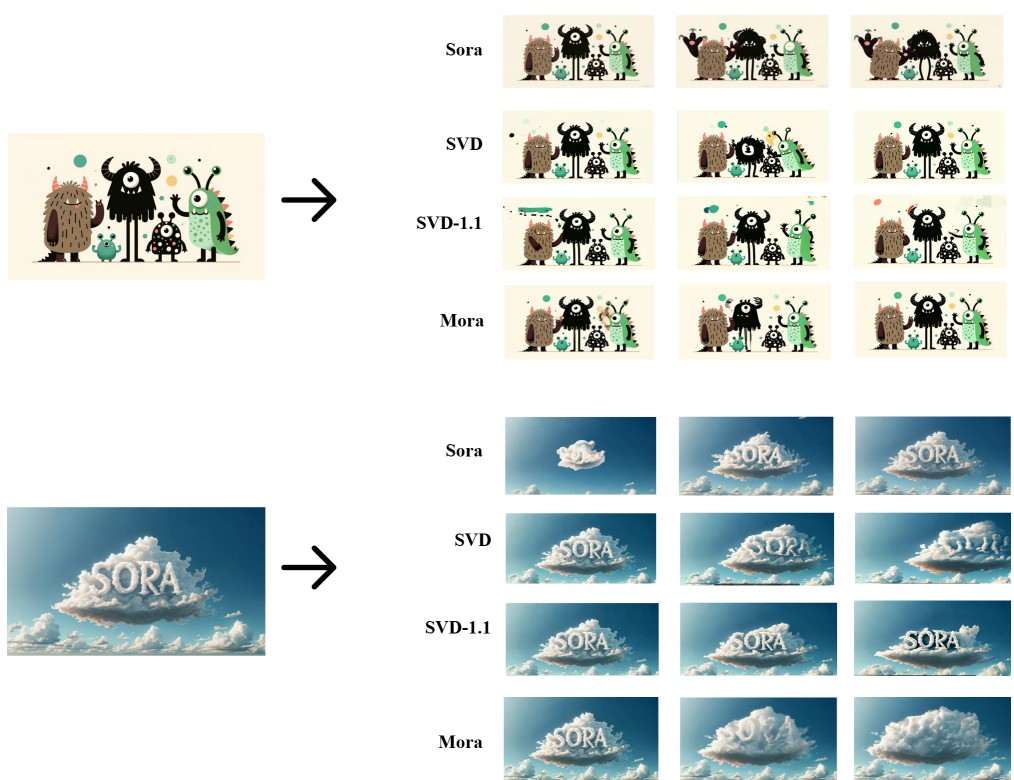

Figure 15: Samples for text-conditional image-to-video generation of Mora and Sora. Prompt for the first line image is: Monster Illustration in flat design style of a diverse family of monsters. The group includes a furry brown monster, a sleek black monster with antennas, a spotted green monster, and a tiny polka-dotted monster, all interacting in a playful environment. The second image's prompt is: An image of a realistic cloud that spells "SORA".

Table 7: Framework comparison capabilities of Sora, Mora and other autonomous agent framework. "✓" indicates the specific feature in the corresponding framework or model. "-" means absence.

| Framework | Role-based agent | SOPs | Human in the loop | Task |
|---|---|---|---|---|
| AutoGPT (Aut) | - | - | - | Code Generation |
| AutoGen (Wu et al., 2023b) | - | ✓ | ✓ | Code Generation |
| MetaGPT (Hong et al., 2023) | ✓ | ✓ | - | Code Generation |
| Sora | - | - | - | Video Generation |
| Mora | ✓ | ✓ | ✓ | Video Generation |

performance, achieving 0.797 in video quality and maintaining the same level of object consistency (0.98) and motion smoothness (0.99) as the full model. However, the overall consistency and quality remain lower than the full Mora model, confirming that learned modulation embeddings contribute significantly to the model's performance. These results demonstrate that each component—human-in-the-loop, self-modulation, and optimized modulation embeddings—plays a crucial role in improving the model's text-to-video generation capabilities. The full Mora model's superior performance validates the importance of our design, providing a clear advantage over the ablated versions.

### B.3 OTHER BENCHMARKS RESULTS

To further demonstrate the effectiveness of our proposed approach and address concerns about overfitting in the data-free training strategy, we conducted extensive evaluations on multiple widely-used benchmarks: T2V-CompBench, EvalCrafter, and VBench. These benchmarks encompass diverse out-of-domain datasets and tasks, providing a comprehensive assessment of the generalization capabilities of our model, Mora. The results are summarized in Table 10.

From the results, as shown in Table 8, Mora outperforms other state-of-the-art models across all three benchmarks, achieving the highest scores in T2V-CompBench (Sun et al., 2024a) (0.5022), EvalCrafter (Liu et al., 2024b) (263), and VBench (Whole Metrics: 0.821). These improvements highlight Mora's ability to generalize well to unseen tasks and datasets, demonstrating the robustness of our multi-agent approach. Compared to Gen2, the second-best performing model, Mora shows a relative improvement of 4.7% in T2V-CompBench, 3.5% in EvalCrafter, and 2.0% in VBench.

These results validate that our multi-agent framework, combined with the self-modulated fine-tuning algorithm, does not merely overfit to the data synthesized by large models. Instead, it leverages the synthesized data and human-in-the-loop preference optimization to enhance inter-agent collaboration, leading to superior performance on challenging out-of-domain scenarios.

Moreover, Mora's performance improvements come without requiring substantial additional computational resources during inference compared to other multi-agent systems. By optimizing inter-agent interactions, Mora achieves these results while maintaining efficient resource utilization.

Table 8: Performance of Mora on out-of-domain test datasets and benchmarks.

| Model | T2V-CompBench | EvalCrafter | VBench (Whole Metrics) |
|---|---|---|---|
| ModelScope | 0.3990 | 218 | 0.756 |
| ZeroScope | 0.3503 | 217 | 0.755 |
| Show-1 | 0.4209 | 229 | 0.789 |
| VideoCrafter2 | 0.4452 | 243 | 0.804 |
| Pika | 0.4306 | 250 | 0.806 |
| Gen2 | 0.4604 | 254 | 0.805 |
| Open-Sora-Plan | 0.4849 | 259 | 0.812 |
| Mora | **0.5022** | **263** | **0.821** |

## B.4 EFFICIENCY OF MULIT-AGENT FRAMEWORK

To address concerns about the efficiency of Mora's multi-agent framework, we provide detailed comparisons of computational requirements, including memory usage and inference time, for various video-related tasks. Table 9 summarizes these results and highlights Mora's ability to achieve state-of-the-art performance while maintaining comparable computational efficiency.

Mora's design prioritizes efficient memory utilization despite the multi-agent nature of the system. Each agent operates independently under the coordination of the SOP, which ensures that memory is released immediately after an agent completes its task. As a result, the maximum memory consumption of the pipeline corresponds only to the agent with the highest memory requirement, which, as shown in Table A, is capped at 46.98 GB for all tasks on an NVIDIA H100 GPU. This efficient memory management enables Mora to handle complex multi-agent operations without exceeding resource constraints.

Inference time is an essential factor when evaluating the efficiency of multi-agent frameworks. For Mora, the most computationally intensive step is the video generation agent, while preceding agents contribute only a negligible portion to the overall time. The results in Table A demonstrate that Mora's inference times are comparable to state-of-the-art models like Open-Sora-Plan and Video-p2p.

The marginal increase in inference time for Mora is well-justified by its substantial gains in video quality across all tasks. Mora consistently achieves higher performance scores, demonstrating that the multi-agent framework, coupled with its collaborative fine-tuning algorithm, provides significant advantages in generating high-quality videos. This trade-off is especially valuable in applications where quality is a priority over minimal inference latency.

## B.5 MULTI-COMPONENT CUMULATIVE ERROR

To address concerns about potential cumulative errors in multi-agent systems, we implemented several architectural and algorithmic techniques to mitigate error accumulation across components. In this section, we present empirical evidence demonstrating how these techniques effectively manage and minimize error propagation, ensuring consistent performance across tasks and video durations.

Table 9: Computational efficiency and performance comparison across different video generation tasks. Task 1: Text-to-Video Generation; Task 2: Image-to-Video Generation; Task 3: Extend Generated Videos; Task 4: Video-to-Video Editing; Task 5: Connect Videos; Task 6: Simulate Digital Worlds. All experiments were conducted on a single NVIDIA H100 GPU.

| Task | Model | Performance | Max Memory Usage | Inference Time |
|---|---|---|---|---|
| 1 | Open-Sora-Plan (T2V) (3s) | 0.765 | 46.98G | 243s |
|   | Mora (3s) | 0.801 | 46.98G | 261s |
| 2 | Open-Sora-Plan (I2V) (3s) | 0.875 | 46.98G | 245s |
|   | Mora (3s) | 0.887 | 46.98G | 253s |
| 3 | Open-Sora-Plan (I2V using original video last frame) (3s) | 0.942 | 46.98G | 245s |
|   | Mora (3s) | 0.983 | 46.98G | 259s |
| 4 | Pika (Proprietary) | 0.232 | - | 143s |
|   | Video-p2p | 0.369 | 26.11G | 208s |
|   | Mora | 0.383 | 46.98G | 253s |
| 5 | Open-Sora-Plan (I2V) | 0.395 | 46.98G | 245s |
|   | Mora | 0.442 | 46.98G | 245s |
| 6 | Open-Sora-Plan (T2V) (3s) | 0.765 | 46.98G | 243s |
|   | Mora (3s) | 0.801 | 46.98G | 261s |

Our self-modulated fine-tuning strategy dynamically adjusts the influence of each agent during the optimization process. This approach ensures that errors introduced by one component do not propagate unchecked to subsequent components. By aligning the outputs of each agent with global system objectives, this technique provides a robust safeguard against error accumulation. The Image-to-Image Agent (A3) serves as a dedicated quality assurance mechanism. It evaluates and refines the outputs of preceding agents, particularly for tasks requiring extended temporal sequences. As shown in Table 10, the absence of A3 leads to significant quality degradation in videos longer than 12 seconds, while its inclusion maintains consistent quality metrics, even for 24-second videos. This demonstrates A3's critical role in mitigating cumulative errors for long-duration tasks. Mora's modular design decouples the operations of individual agents, reducing interdependencies and minimizing error propagation. Ablation studies (Table 11) reveal that the removal of specific components, such as A1 (human input) and A3, impacts performance but does not lead to catastrophic failure. This highlights the robustness of Mora's design in isolating errors within specific modules.

Table 10 compares video quality scores for Mora with and without the Image-to-Image Agent (A3) across different video durations. The results show a clear degradation in quality when A3 is removed, particularly for longer videos. For example, at 24 seconds, Mora without A3 scores only 0.012, compared to 0.773 with A3, emphasizing the agent's importance in maintaining quality over time. Similarly, Table 11 illustrates the impact of removing individual agents on performance. While the removal of A1 (human input) or A2 causes minor fluctuations in quality, the removal of A3 results in a noticeable drop, especially for longer videos. This demonstrates that while the system is resilient to partial component failure, A3 is critical for maintaining consistent performance across complex tasks.

Table 10: Comparison of video quality scores between Mora with and without the Image-to-Image Agent (A3) across different video durations. The scores range from 0 to 1, with higher values indicating better quality. Results show that A3 is crucial for maintaining consistent quality in longer videos, particularly beyond 12 seconds.

| Method | 3s | 6s | 12s | 15s | 18s | 21s | 24s |
|---|---|---|---|---|---|---|---|
| Mora (w/o A3) | 0.800 | 0.794 | 0.780 | 0.622 | 0.439 | 0.211 | 0.012 |
| Mora | 0.800 | 0.799 | 0.800 | 0.784 | 0.775 | 0.773 | 0.773 |

## B.6 FRAMEWORK FOR DIFFERENT AGENTS

To investigate the impact of substituting Mora's original agents (A1–A5) with alternative models, we conducted extensive ablation studies across various tasks. Table 12 summarizes the results, showcasing the performance of different agent configurations compared to Mora's default setup. The findings reveal the importance of each component and the synergistic advantages of Mora's carefully curated multi-agent framework.

**A1: Text Enhancer** Replacing Mora's original text enhancer with alternatives such as Claude 3.5 or LLaMA 3.1 (70B and 8B) demonstrates relatively minor performance degradation across most

Table 11: Video quality scores across different configurations, demonstrating the impact of the Image-to-Image Agent (A3) and human input (A1, A2) in maintaining consistent quality. Scores range from 0 to 1, with higher values indicating better quality.

| Method | Performance |
|---|---|
| Mora (w/o A1 (Human Input)) | 0.801 |
| Mora (w/o A2 (Human Input)) | 0.809 |
| Mora (w/o A1 & A2 (Human Input)) | 0.809 |
| Mora (w/o A3) | 0.780 |
| Mora (w/o A1 & A2 (Human Input) and w/o A3) | 0.781 |

tasks. The results suggest that A1's role is less critical for Mora's overall performance, as alternative models maintain comparable quality in most metrics. However, the smaller-scale LLaMA 3.1 (8B) model causes noticeable degradation in image quality (IQ drops from 0.399 to 0.333), indicating that model size and capability can influence specific metrics.

**A2: Text-to-Image Model** The Text-to-Image agent (A2) exhibits greater sensitivity to model replacement. Substituting SDXL v1.1 with SD 1.5 results in a substantial decline in Task 6 image quality (IQ drops from 0.398 to 0.288). This highlights the importance of advanced text-to-image generation capabilities for maintaining high-quality visuals, particularly in scenarios requiring detailed imagery.

**A3: Image Editing Model** The Image Refinement agent (A3) plays a crucial role in Mora's pipeline. Replacing SDXL v1.1 Refiner with InstructPix2Pix leads to significant performance degradation, particularly in temporal consistency (TCON drops from 0.983 to 0.950) and image quality (IQ drops from 0.399 to 0.281). These results underscore A3's importance in ensuring visual consistency and quality across generated frames.

**A4: Image-to-Video Model** The Video Generation agent (A4) emerges as the cornerstone of Mora's framework. Replacing A4 with Open-Sora v1.2 results in dramatic performance losses across all tasks, with the most notable decline in Task 3 temporal consistency (TCON drops from 0.983 to 0.723) and Task 2 average performance (0.887 to 0.704). These findings validate A4's critical role in maintaining temporal coherence and overall video quality.

**A5: Video Transition Model** Mora's Video Transition agent (A5) is relatively robust to replacement. Substituting it with Open-Sora v1.2 or SEINE causes only minimal variations in performance, such as a small drop in Task 5 average performance (from 0.442 to 0.435 with SEINE). These results suggest that A5 contributes to Mora's performance but is less sensitive to the choice of model compared to other agents.

These findings validate Mora's original agent selection, demonstrating that its configuration achieves the optimal balance between performance and quality. The consistent performance degradation observed when substituting agents underscores the effectiveness of Mora's integrated multi-agent architecture and self-modulated fine-tuning strategy.

### B.7 AGENT SUCCESS RATE

To provide a quantitative analysis of agent success rates and justify the benefits of Mora's multi-agent framework, we evaluate the proportion of tasks successfully completed by agents without requiring external corrections or reinitialization. This metric, termed the agent success rate, reflects the robustness and efficiency of the multi-agent system. Table 13 compares Mora's performance and agent success rates with and without the self-modulated fine-tuning algorithm. The results show a significant improvement, with Mora's agent success rate increasing from 34.5% to 91.5% and its performance score rising from 0.776 to 0.800. These findings demonstrate that self-modulated fine-tuning not only optimizes agent collaboration but also enhances their ability to autonomously and effectively complete tasks.

The high agent success rate achieved with self-modulated fine-tuning highlights the effectiveness of Mora's multi-agent system. By dynamically adjusting the influence of each agent during the optimization process, the fine-tuning approach ensures that agents are better aligned and operate in harmony, reducing conflicts and dependencies on external interventions. The enhanced success rate

Table 12: The quality performance of different ablation versions of Mora with different agent configurations (A1-A5). Task 1-6 represent different evaluation metrics as described in the paper. Scores are bolded for Mora's original configuration to emphasize its superior performance.

| Model | Task 1 (Avg.) | Task 2 (Avg.) | Task 3 (TCON) | Task 4 (Avg.) | Task 5 (Avg.) | Task 6 (IQ) |
|---|---|---|---|---|---|---|
| Mora (ours) | **0.800** | **0.887** | **0.983** | **0.383** | **0.442** | **0.399** |
| A1: Different Text Enhancer | | | | | | |
| Claude 3.5 (Anthropic, 2024) | 0.800 | 0.887 | 0.983 | 0.383 | 0.442 | 0.397 |
| LLaMA 3.1 70B (Meta AI, 2024) | 0.799 | 0.887 | 0.983 | 0.384 | 0.442 | 0.353 |
| LLaMA 3.1 8B (Meta AI, 2024) | 0.797 | 0.885 | 0.983 | 0.381 | 0.442 | 0.333 |
| A2: Different Text-to-Image Model | | | | | | |
| SDXL v1.1 | 0.787 | 0.887 | 0.983 | 0.381 | 0.442 | 0.398 |
| SD 1.5 (Rombach et al., 2022a) | 0.783 | 0.887 | 0.983 | 0.381 | 0.442 | 0.288 |
| A3: Different Image Editing Model | | | | | | |
| SDXL v1.1 Refiner | 0.799 | 0.877 | 0.954 | 0.383 | 0.442 | 0.393 |
| InstructPix2Pix | 0.787 | 0.804 | 0.950 | 0.383 | 0.442 | 0.281 |
| A4: Different Image-to-Video Model | | | | | | |
| Open-Sora v1.2 | 0.770 | 0.704 | 0.723 | 0.329 | 0.442 | 0.384 |
| A5: Different Video Transition Model | | | | | | |
| Open-Sora v1.2 | 0.800 | 0.887 | 0.983 | 0.383 | 0.422 | 0.399 |
| SEINE | 0.800 | 0.887 | 0.983 | 0.383 | 0.435 | 0.399 |

also underscores the system's efficiency, as fewer task failures result in smoother operations and faster execution. These results validate Mora's multi-agent design and its ability to handle complex video generation tasks efficiently, reinforcing its suitability for real-world applications.

Table 13: Performance and agent success rates of Mora with and without self-modulated fine-tuning.

| Method | Performance | Agent Success Rate |
|---|---|---|
| Without self-modulated fine-tuning | 0.776 | 34.5% |
| **With self-modulated fine-tuning** | **0.800** | **91.5%** |

## C  CAPABILITIES ANALYSIS

Compared to the closed-source baseline model Sora and autonomous agents such as MetaGPT (Hong et al., 2023), our framework, Mora, offers enhanced functionalities for video generation tasks. As shown in Table 7, Mora encompasses a comprehensive range of capabilities designed to handle diverse and specialized video generation tasks effectively. The integration of Standard Operating Procedures (SOPs) such as role-play expertise, structured communication, and streamlined workflows, along with human-in-the-loop systems, significantly refines the control and quality of video generation. While other baseline methods can adapt SOP-like designs to boost their performance, they typically do so only within the realm of code generation tasks. In contrast, models like Sora lack the capability for fine-grained control or autonomous video generation, highlighting the advanced capabilities of Mora in this domain.

A detailed comparison of tasks between Mora and Sora is presented in Table 14. This comparison demonstrates that, through the collaboration of multiple agents, Mora is capable of accomplishing the video-related tasks that Sora can undertake. This comparison highlights Mora's adaptability and proficiency in addressing a multitude of video generation challenges.

Table 14: Task comparison between Sora, Mora and other existing models.

| Tasks | Example | Sora | Mora | Others |
|---|---|---|---|---|
| Text-to-video Generation | Tᴛ → ▶ | ✓ | ✓ | (Girdhar et al., 2023; Wang et al., 2023a; Chen et al., 2024a; Ma et al., 2024b) |
| Image-to-Video Generation | Tᴛ ⊕ 🖼 → ▶ | ✓ | ✓ | (Blattmann et al., 2023; pik; Gen, a) |
| Extend Generated Videos | ▶ → ▶▶ | ✓ | ✓ | - |
| Video-to-Video Editing | Tᴛ ⊕ ▶ → ▶ | ✓ | ✓ | (Molad et al., 2023; Liew et al., 2023; Ceylan et al., 2023) |
| Connect Videos | ▶ ⊕ ▶ → ▶ | ✓ | ✓ | (Chen et al., 2023c) |
| Simulate Digital Worlds | Tᴛ → 🌐 | ✓ | ✓ | - |

## D Discussion

**Advantages of Mora.** Mora introduces a groundbreaking multi-agent framework for video generation, advancing the field by enabling a variety of tasks such as text-to-video conversion and digital world simulation. Unlike closed-source counterparts like Sora, Mora's open framework offers seamless integration of various models, enhancing flexibility and efficiency for diverse applications. As an open-source project, Mora significantly contributes to the AI community by democratizing access to advanced video generation technologies and fostering collaboration and innovation. Future research is encouraged to improve the framework's efficiency, reduce computational demands, and explore new agent configurations to enhance performance.

**Limitations of Mora.** Mora faces significant limitations, including challenges in collecting high-quality video datasets due to copyright restrictions, resulting in difficulties in generating lifelike human movements. Its video quality and length capabilities fall short compared to Sora, with noticeable degradation beyond 12 seconds. Mora also struggles with interpreting and rendering motion dynamics from prompts, lacking control over specific directions. Furthermore, the absence of human labeling information in video datasets leads to results that may not align with human visual preferences, highlighting the need for datasets that adhere to physical laws.

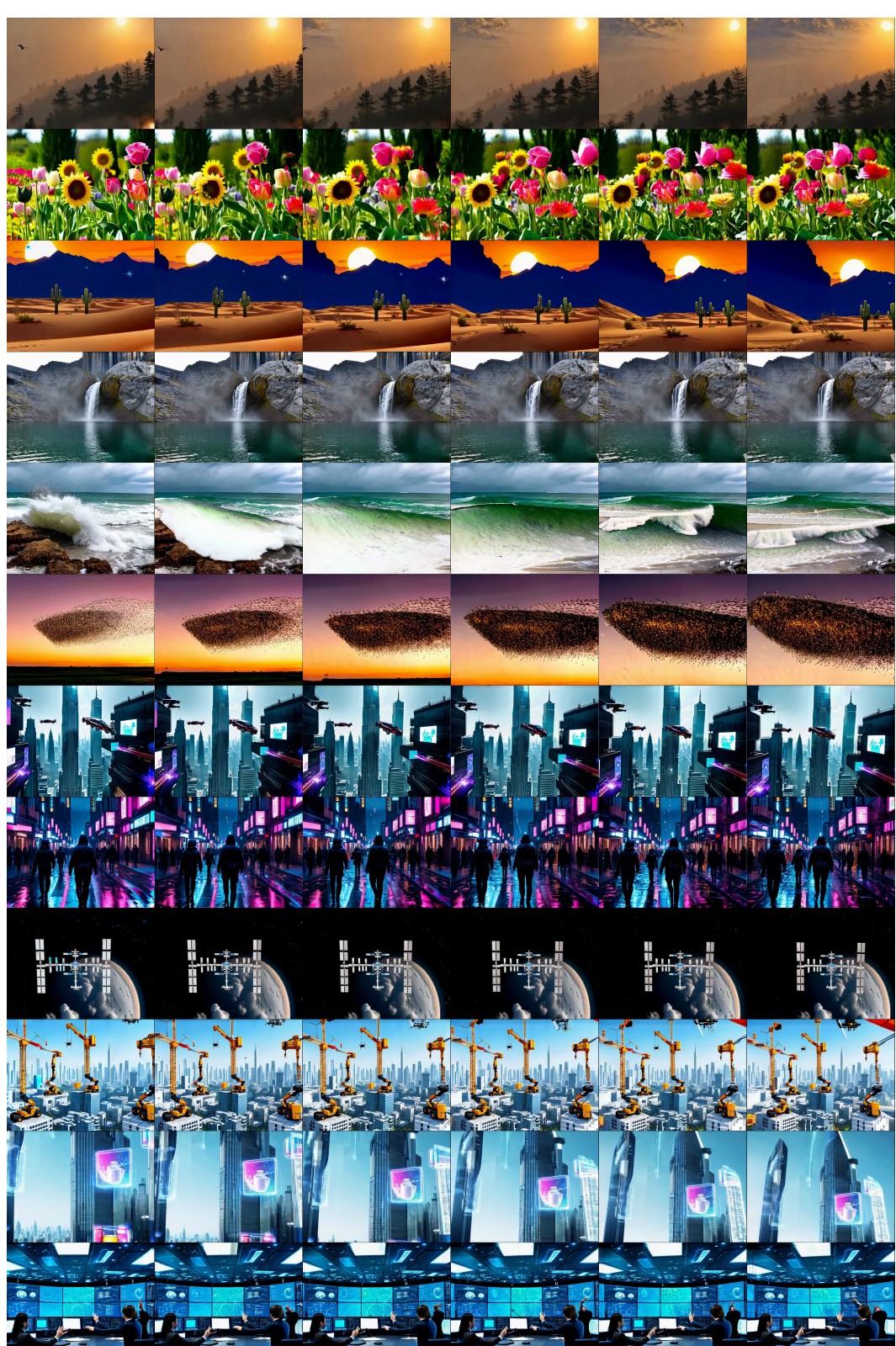

Figure 16: Some video examples generated by Mora.

