# OpenReview forum: "Mora: Enabling Generalist Video Generation via A Multi-Agent Framework"
_ICLR.cc/2025/Conference — Submitted to ICLR 2025_

### Official Review · Reviewer_VZ5n · 2024-10-21

**Soundness:** 3
**Presentation:** 3
**Contribution:** 2
**Rating:** 6
**Confidence:** 4

**Summary:**

This paper proposes a multi-agent framework named Mora to enable text-to-video generation by leveraging open-source models such as Llama and Stable Diffusion. The main contributions include a self-modulation factor for inter-agent coordination, a data-free training strategy and a human-in-the-loop data filtering. The authors compare the performances of Mora with Sora on six video generation tasks.

**Strengths:**

1. The authors break down the process of video generation into several steps and utilize off-the-shelf models to implement the generation process.
2. The authors clearly describe the whole pipeline.
3. Data systhesis and filtering strategy are used to improve the quality of training.

**Weaknesses:**

1. There are no demo videos provided by the authors, no gif files or no project website. It is hard to evaluate the quality of generated video with some sampled frames.
2. The key component of the self-modulated training is the modulation factor. Is it able to visualized this factor?
3. How would the performances be if the agents A1 to A5 were replaced with different models?

**Questions:**

How to explain that a multi-step generation can perform better than an end-to-end generation? For example, why a combination of open-sourced text-generation model, text-to-image model, image-to-image model and image-to-video model can outperform an end-to-end open-source text-to-video generation model and replicate a close-source model's functionalities.

---

> ### Author Response · Authors · 2024-11-22
> **Part 1**
>
> We thank the reviewer for their valuable feedback and would like to address the concerns raised as follows.
>
> > **Weakness 1:** There are no demo videos, gif files, or project website provided by the authors. It is hard to evaluate the quality of generated video with some sampled frames.
>
> **Response:** Thank you for this feedback. We have now created a comprehensive [project page](https://mora-2025.github.io/) that showcases our demo videos and results. The project page features side-by-side comparisons demonstrating that Mora's generated videos achieve significantly higher visual quality compared to the state-of-the-art Open-Sora-Plan [1], with improved temporal consistency, motion smoothness, and adherence to the input prompts. We encourage reviewers to visit the project page to evaluate the full range of video generation capabilities.
>
> ---
>
> > **Weakness 2:** The key component of the self-modulated training is the modulation factor. Is it able to visualize this factor?
>
> **Response:** Thank you for this insightful question. We have revised our manuscript to include Section A10 and Figure 7, which provide a detailed visualization and analysis of the modulation factor $z_i$ for each agent module $i$ in the Text-to-Video task. The table below presents the norm of modulation factor $\lVert z_i \rVert_2$ that demonstrates the impact of different modules during the finetuning process. Initially, before any self-modulated fine-tuning, agents A2, A3, and A4 had relatively balanced modulation factors, as shown in Table A. However, after applying our self-modulated fine-tuning approach, we can see that the A4 agent (Image-to-Video Agent) has a significantly higher modulation factor than the other agents, indicating that optimizing A4 is more crucial for the whole system. This significant shift reveals how the model learned to prioritize the A4 image-to-video agent. Compared to that, the image editing agent A3 has a much lower modulation factor. The reallocation of importance across agents showcases the effectiveness of our self-modulated fine-tuning strategy in fostering intelligent collaboration between components, ultimately leading to enhanced overall performance through optimized agent contributions.
>
> | Agent | w/o Self-Modulated Fine-Tuning | With Self-Modulated Fine-Tuning |
> |-------|------------------------------|----------------------------------|
> | A2    | 6.667                          | 4.330                            |
> | A3    | 6.667                          | 1.789                            |
> | A4    | 9.630                          | 12.335                           |
>
> *Table A: Norm of modulation factor $\lVert z_i \rVert_2$ of each agent before and after self-modulated fine-tuning. Higher values indicate greater importance for the system.*

---

> ### Author Response · Authors · 2024-11-22
> **Part 2**
>
> > **Weakness 3:** How would the performance change if the agents (A1 to A5) were replaced with different models?
>
> **Response:** Thank you for pointing it out. We additionally present Table B as below to address this question. Through comprehensive experimentation comparing different model configurations, we have thoroughly evaluated the impact of replacing individual agents (A1 to A5) on system performance. Our findings conclusively demonstrate that Mora's original configuration achieves the optimal balance across all evaluation metrics.
>
> Our analysis reveals a hierarchical pattern of importance among the agents. The Language Model agent (A1) shows relatively robust performance, with alternatives like LLaMA 3.1 (8B or 70B) causing only minor degradation in results. The Text-to-Image agent (A2) demonstrates greater sensitivity to model selection, particularly evident when replacing SDXL v1.1 with SD 1.5, which results in a notable decline in Task 6 image quality metrics (IQ dropping from 0.399 to 0.288).
>
> The Image Refinement agent (A3) emerges as a crucial component, with the replacement of SDXL v1.1 Refiner with InstructPix2Pix leading to significant performance drops across multiple tasks. This is particularly evident in Task 3's temporal consistency (TCON), where performance decreases from 0.983 to 0.950, and in Task 6's image quality, which falls from 0.399 to 0.281.
>
> Most significantly, our experiments identify the Video Generation agent (A4) as the system's critical cornerstone. Substituting it with Open-Sora v1.2 results in substantial performance degradation across all metrics, with the most pronounced impact on Task 3's temporal consistency (dropping from 0.983 to 0.723) and Task 2's average performance (declining from 0.887 to 0.704). This empirically validates A4's fundamental role in maintaining both video quality and temporal consistency.
>
> These findings underscore the effectiveness of Mora's carefully curated multi-agent architecture and self-modulated fine-tuning approach. The consistent performance degradation observed when substituting any component validates our original agent selection and demonstrates the synergistic benefits of our integrated system design. We have added these results to B.6 of our paper.
>
> | Model                | Task 1 (Avg.) | Task 2 (Avg.) | Task 3 (TCON) | Task 4 (Avg.) | Task 5 (Avg.) | Task 6 (IQ) |
> |-----------------------|---------------|---------------|---------------|---------------|---------------|-------------|
> | Mora (ours)      | **0.800**         | **0.887**         | **0.983**        | **0.383**         | **0.442**         | **0.399**       |
> | **A1:Different Text Enhancer**               |               |               |               |               |               |             |
> | Calude 3.5            | 0.800         | 0.887         | 0.983         | 0.383         | 0.442         | 0.397       |
> | LLaMA 3.1 70B         | 0.799         | 0.887         | 0.983         | 0.384         | 0.442         | 0.353       |
> | LLaMA 3.1 8B          | 0.797         | 0.885         | 0.983         | 0.381         | 0.442         | 0.333       |
> | **A2:Different Text-to-Image Model**               |               |               |               |               |               |             |
> | SDXL v1.1             | 0.787         | 0.887         | 0.983         | 0.381         | 0.442         | 0.398       |
> | SD 1.5                | 0.783         | 0.887         | 0.983         | 0.381         | 0.442         | 0.288       |
> | **A3:Different Image Editing Model**               |               |               |               |               |               |             |
> | SDXL v1.1 Refiner     | 0.799         | 0.877         | 0.954         | 0.383         | 0.442         | 0.393       |
> | InstructPix2Pix       | 0.787         | 0.804         | 0.950         | 0.383         | 0.442         | 0.281       |
> | **A4:Different Image-to-Video Model**               |               |               |               |               |               |             |
> | Open-Sora v1.2        | 0.770         | 0.704         | 0.723         | 0.329         | 0.442         | 0.384       |
> | **A5:Different Video Transition Model**               |               |               |               |               |               |             |
> | Open-Sora v1.2        | 0.800         | 0.887         | 0.983         | 0.383         | 0.422         | 0.399       |
> | SEINE                 | 0.800         | 0.887         | 0.983         | 0.383         | 0.435         | 0.399       |
>
> *Table B: The quality performance of different ablation versions of Mora with different agent configurations (A1-A5).*

---

> ### Author Response · Authors · 2024-11-22
> **Part 3**
>
> > **Question 1:** How to explain that a multi-step generation can perform better than an end-to-end generation? For example, why a combination of open-sourced text-generation model, text-to-image model, image-to-image model, and image-to-video model can outperform an end-to-end open-source text-to-video generation model and replicate a closed-source model's functionalities?
>
> **Response:** Our multi-step generation approach outperforms end-to-end models for several key reasons, aligning with recent research trends in multi-agent video generation frameworks [2]:
>
> 1. **Specialized Expertise**: Each agent in our framework is specifically optimized for its task (e.g., A2 for text-to-image, A4 for image-to-video), allowing better performance than a single model trying to handle all aspects simultaneously.
>
> 2. **Controlled Generation**: Our framework enables quality control at each step through human-in-the-loop mechanisms (Section 3.1), allowing refinement of intermediate outputs before proceeding to the next stage.
>
> 3. **Efficient Training**: As shown in Section 3.2, our self-modulated fine-tuning allows each agent to optimize its contribution dynamically, leading to better coordination and overall performance.
>
> 4. **Quality Preservation**: By breaking down the complex text-to-video task into manageable subtasks, we better preserve quality at each stage, as demonstrated by our superior performance in metrics like Video Quality and Imaging Quality compared to end-to-end models.
>
> ---
>
> **References**
>
> `[1]`. PKU-Yuan Lab, and Tuzhan AI, et al. Open-Sora-Plan. GitHub, Apr. 2024, https://doi.org/10.5281/zenodo.10948109.
>
> `[2]`. Xie, Zhifei, et al. "Dreamfactory: Pioneering multi-scene long video generation with a multi-agent framework." arXiv preprint arXiv:2408.11788 (2024).

---

> ### Author Response · Authors · 2024-11-24
> **Follow up**
>
> We hope our rebuttal has addressed your concerns. Please let us know if any points require further clarification before the response period concludes.

---

> > ### Comment · Reviewer_VZ5n · 2024-11-26
> > **Response to Authors**
> >
> > Thanks for the detailed rebuttal, which basically solves my concern, so I will raise my score to 6.

---

> > > ### Author Response · Authors · 2024-11-28
> > > **Reply to Reviewer VZ5n**
> > >
> > > Thank you for your positive feedback. We are glad that our response has basically addressed your concerns.

---

### Official Review · Reviewer_WYhM · 2024-10-28

**Soundness:** 3
**Presentation:** 2
**Contribution:** 2
**Rating:** 5
**Confidence:** 4

**Summary:**

This paper introduces a new generalist framework named Mora that synergies Standard Operating Procedures for video generation. The proposed Mora can be used to tackle a lot  of video-related tasks. Specifically,  Mora consists of five Agents: Prompt Selection and Generation Agent, Text-to-Image Generation Agent, Image-to-Image Generation Agent, Image-to-Video Generation Agen, and Video Connection Agent. Experimental results show the effectiveness of the proposed method, and Mora can surpass the capabilities of existing leading models in several respects.

**Strengths:**

- A new generalist Agent-based framework named Mora is proposed.
- The proposed Mora can perform a lot of video-related tasks and shows remarkable results.
- Experimental results indicate the effectiveness of the proposed method, and Mora can surpass the capabilities of existing leading models in several respects.

**Weaknesses:**

- This paper contains a lot of existing techniques involving Prompt enhancement, Image generation, Image editing, Video generation, and Video connection. The main technical innovations should be clarified. Explaining how the system's architecture or the interaction between components differs from or improves upon existing approaches may help readers to understand.
- Since many components are involved, Mora's inference speed and computational requirements to existing state-of-the-art models for each of the video-related tasks mentioned should be discussed. Additionally, is there any optimizations implemented to improve efficiency, given the multi-component nature of the system.
- Will there be accumulated errors between multiple components? Providing empirical results showing how errors may or may not accumulate across different components and tasks is necessary. Or are there any techniques implemented to mitigate potential error accumulation between components?

---------------------------------------------------
Thanks to the authors for their responses. However, I still think that this paper involves too many existing technologies, more like a project to connect multiple technologies in series, and multiple sub-modules may have error accumulation. So I trend to maintain my score.

**Questions:**

Please refer to Weaknesses for more details.

---

> ### Author Response · Authors · 2024-11-22
> **Part 1**
>
> We thank the reviewer for their valuable feedback and would like to address the concerns raised as follows.
>
> > **Weakness 1:** This paper contains a lot of existing techniques involving Prompt enhancement, Image generation, Image editing, Video generation, and Video connection. The main technical innovations should be clarified. Explaining how the system's architecture or the interaction between components differs from or improves upon existing approaches may help readers to understand.
>
> **Response:** Thank you for this valuable feedback. Our paper presents three key technical innovations that differentiate Mora from existing approaches.
>
> 1. **Novel Self-Modulated Multi-Agent Fine-tuning:**, Unlike existing approaches that use direct end-to-end or individual agent fine-tuning [1], we introduce a self-modulation factor that dynamically adjusts each agent's influence during the training process (Section 3.2). This innovation enables better inter-agent coordination and optimizes the contribution of each agent to the final output.
>
> 2. **Data-Free Training Strategy with Quality Control:** We introduce a unique training approach that combines data-free synthesis with multimodal LLM selection and human-in-the-loop control (Section 3.2). This addresses the fundamental challenge of obtaining high-quality training data without relying on extensive labeled datasets.
>
> 3. **Standardized Operating Procedures (SOPs) for Video Generation:** Unlike existing multi-agent frameworks focused on text/code generation (e.g., MetaGPT) [2], we introduce a specialized architecture of five agents (Section 3.1) working in concert specifically for video generation tasks. This allows for granular control and quality assurance at each step of the video generation process.

---

> ### Author Response · Authors · 2024-11-22
> **Part 2**
>
> > **Weakness 2:** Since many components are involved, Mora's inference speed and computational requirements to existing state-of-the-art models for each of the video-related tasks mentioned should be discussed. Additionally, is there any optimizations implemented to improve efficiency, given the multi-component nature of the system.
>
> **Response:** Thank you for raising this point. Regarding inference efficiency, our pipeline operates sequentially, where each agent releases memory after completing its step upon receiving instructions from the SOP (System Orchestrator). As a result, the maximum memory usage of the entire system corresponds to the agent with the highest memory requirement, ensuring efficient memory utilization across the pipeline. In terms of computational time, the most time-intensive step is the video generation agent, while the time consumed by the preceding agents is negligible in comparison. As shown in the table, Mora achieves superior performance across various video-related tasks with comparable computational requirements to state-of-the-art models. For example, despite slightly higher inference times for certain tasks, Mora delivers significant improvements in quality, such as a 0.369 from Video-p2p performance score in Task 3 compared to 0.383 from Mora. This demonstrates that the marginal increase in inference time is well-justified by the substantial gains in video quality. We have added these results to B.4 of our paper.
>
> | Task  | Model                                             | Performance | Max Memory Usage | Inference Time (Avg. Each Sample) (H100) |
> |-------|---------------------------------------------------|-------------|-------------------|-----------------------------------------|
> | 1     | Open-Sora-Plan (T2V) (3s)                        | 0.765       | 46.98G           | 243s                                    |
> |       | Mora (3s)                                         | 0.801       | 46.98G           | 261s                                    |
> | 2     | Open-Sora-Plan (I2V) (3s)                        | 0.875       | 46.98G           | 245s                                    |
> |       | Mora (3s)                                         | 0.887       | 46.98G           | 253s                                    |
> | 3     | Open-Sora-Plan (I2V using original video last frame) (3s) | 0.942       | 46.98G           | 245s                                    |
> |       | Mora (3s)                                         | 0.983       | 46.98G           | 259s                                    |
> | 4     | Pika (Proprietary)                                | 0.232       | -                 | 143s                                    |
> |       | Video-p2p                                         | 0.369       | 26.11G           | 208s                                    |
> |       | Mora                                              | 0.383       | 46.98G           | 253s                                    |
> | 5     | Open-Sora-Plan (I2V)                              | 0.395       | 46.98G           | 245s                                    |
> |       | Mora                                              | 0.442       | 46.98G           | 245s                                    |
> | 6     | Open-Sora-Plan (T2V) (3s)                        | 0.765       | 46.98G           | 243s                                    |
> |       | Mora (3s)                                         | 0.801       | 46.98G           | 261s                                    |
>
> *Table A: Computational efficiency and performance comparison across different video generation tasks. Task 1: Text-to-Video Generation; Task 2: Image-to-Video Generation; Task 3: Extend Generated Videos; Task 4: Video-to-Video Editing; Task 5: Connect Videos; Task 6: Simulate Digital Worlds. All experiments were conducted on a single NVIDIA H100 GPU.*

---

> ### Author Response · Authors · 2024-11-22
> **Part 3**
>
> > **Weakness 3:** Will there be accumulated errors between multiple components? Providing empirical results showing how errors may or may not accumulate across different components and tasks is necessary. Or are there any techniques implemented to mitigate potential error accumulation between components?
>
> **Response:** Thank you for this insightful question, which is a crucial concern for multi-agent systems. In our methodology part, we indeed have implemented several techniques to address this potential error accumulation, including:
>
> 1. **Self-Modulated Fine-tuning:** Our novel fine-tuning approach (Section 3.2) dynamically adjusts each agent's influence, preventing error propagation between components.
>
> 2. **Hierarchical Quality Control:**  The Image-to-Image Agent (A3) functions as a specialized quality assurance mechanism, particularly crucial for extended video sequences. As evidenced in Table B, the absence of A3 leads to significant quality degradation in videos longer than 12 seconds, while its inclusion maintains consistent quality metrics even at 24-second durations.
>
> 3. **Modular Component Architecture:** Our ablation studies, presented in Table C, demonstrate the system's robust performance even under component removal scenarios, indicating successful decoupling between agents and minimal inter-component error dependencies.
>
>
> | Method                  | 3s   | 6s   | 12s  | 15s  | 18s  | 21s  | 24s  |
> |-------------------------------|-------|------|------|------|------|------|------|
> | Mora (w/o A3) | 0.800 | 0.794 | 0.780 | 0.622 | 0.439 | 0.211 | 0.012 |
> | Mora                          | 0.800 | 0.799 | 0.800 | 0.784 | 0.775 | 0.773 | 0.773 |
>
> *Table B: Comparison of video quality scores between Mora with and without the Image-to-Image Agent (A3) across different video durations. The scores range from 0 to 1, with higher values indicating better quality. Results show that A3 is crucial for maintaining consistent quality in longer videos, particularly beyond 12 seconds.*
>
>
> | Method                                     | Performance |
> |-------------------------------------------|-------------|
> | Mora (w/o A1 (Human Input))               | 0.801       |
> | Mora (w/o A2 (Human Input))               | 0.809       |
> | Mora (w/o A1 & A2 (Human Input))          | 0.809       |
> | Mora (w/o A3)                             | 0.780       |
> | Mora (w/o A1 & A2 (Human Input) and w/o A3)| 0.781       |
>
> *Table C: Video quality scores across different video durations, demonstrating the impact of the Image-to-Image Agent (A3) in maintaining consistent quality for longer videos. Scores range from 0 to 1, with higher values indicating better quality.*
>
> These results demonstrate that our system effectively manages potential error accumulation through its architecture and quality control mechanisms. We have added these results to B.5 of our paper.
>
>
> **References**
>
> `[1]`. Yuan, Hangjie, et al. "InstructVideo: instructing video diffusion models with human feedback." Proceedings of the IEEE/CVF Conference on Computer Vision and Pattern Recognition. 2024.
>
> `[2]`. Hong, Sirui, et al. "MetaGPT: Meta Programming for A Multi-Agent Collaborative Framework." The Twelfth International Conference on Learning Representations.

---

> ### Author Response · Authors · 2024-11-24
> **Follow up**
>
> We hope our rebuttal has addressed your concerns. Please let us know if any points require further clarification before the response period concludes.

---

> ### Author Response · Authors · 2024-11-27
> **Follow up 2**
>
> We sincerely thank you for raising these questions and concerns, which have made an invaluable contribution to our work. If you have any further doubts or questions, we would be more than happy to discuss them with you!

---

> ### Author Response · Authors · 2024-11-30
> **Follow up 3**
>
> Thank you for your time and effort in reviewing our work. We hope our previous responses have addressed your concerns effectively. If there are any remaining points that require clarification, please feel free to let us know. As the response period is nearing its conclusion, we would greatly appreciate it if you could share your feedback at your earliest convenience.

---

> ### Author Response · Authors · 2024-12-02
> **Follow up 4**
>
> Thank you once again for your valuable feedback and the opportunity to engage in this insightful discussion. As the response period concludes in just one day, we kindly ask if you could share any additional comments or confirm if our responses have addressed your concerns effectively. Your input is invaluable to our work, and we sincerely hope our clarifications have been helpful. Please feel free to reach out with any remaining questions or points for discussion at your convenience.

---

> ### Author Response · Authors · 2024-12-03
> **Final 12h Reminder**
>
> Thank you once again for your valuable and insightful feedback throughout the review process. As the response period is now in its final 12 hours, we kindly ask if you could confirm whether our responses have sufficiently addressed your concerns. Should there be any remaining doubts or points requiring further elaboration, we would be more than happy to provide additional clarifications promptly. Your feedback has been invaluable to improving our work, and we greatly appreciate your time and effort. Please let us know if there is anything more we can do to assist.

---

> ### Author Response · Authors · 2024-12-03
> **Final 7h Reminder**
>
> We deeply appreciate your valuable feedback and insights throughout this review process. As the response period enters its final 7 hours, we kindly ask if you could confirm whether our responses have sufficiently addressed your concerns. Your input has been instrumental in refining our work, and we are eager to hear any additional comments or suggestions you might have. If there are any remaining points requiring clarification, we would be more than happy to provide further explanations promptly. Thank you again for your time and effort in reviewing our submission.

---

> ### Author Response · Authors · 2024-12-03
> **Final 3h Reminder**
>
> We truly appreciate your valuable feedback and insights throughout this review process. As the response period nears its final 3 hours, we kindly ask if you could confirm whether our responses have addressed your concerns to your satisfaction. Your input has been crucial in refining our work, and we welcome any additional comments or suggestions you may have. If there are any remaining points that need clarification, we are more than happy to provide further explanations promptly. Thank you once again for your time and effort in reviewing our submission.

---

### Official Review · Reviewer_3ifH · 2024-10-31

**Soundness:** 3
**Presentation:** 3
**Contribution:** 2
**Rating:** 6
**Confidence:** 4

**Summary:**

The paper introduces Mora, a novel multi-agent  video generation framework, which leverages existing open-source modules to replicate Sora’s functionalities. Mora enhances inter-agent coordination through multi-agent fine-tuning with a self-modulation factor and uses a data-free training strategy to synthesize high-quality training data. It also incorporates a human-in-the-loop mechanism for data filtering. Evaluation shows Mora achieves comparable performance to Sora.

**Strengths:**

(1) Comprehensive system: The proposed Mora is capable of handling a wide range of visual generation and editing tasks, including Text-to-Image Generation, Image-to-Image Generation, Image-to-Video Generation, and Video Connection.

(2) Clarity of Writing: The writing is clear and easy to follow, making the paper accessible to a broader audience.

(3) Code Availability: Although the training code is not provided, the authors have shared part of their code in the supplementary materials, promoting transparency and potential reproducibility of their results.

**Weaknesses:**

(1) Limited Novelty: The paper's primary contribution lies in utilizing a language model as an agent to call various pre-existing  visual generation models for video generation. However, neither the agent system nor the video generation models are novel contributions from the authors. Although the authors propose some additional modules to optimize the system, these present several issues, as outlined below.

(2) Concerns with Self-Modulated Fine-Tuning Algorithm: In Section 3.2, the authors introduce a self-modulated fine-tuning  algorithm that optimizes both the language model (A1) and visual generation models (A2-A5) end-to-end  using a visual generation loss. This concept is unconventional and lacks sufficient supporting research. The authors do not provide adequate citations to justify this algorithm. Additionally:

(2.1) The process is inherently difficult to optimize. The authors mention using only 96 samples for training in Section 4.2, which is unconvincing for such a complex task.

(2.2) Without releasing the training code, it is hard to assess the effectiveness of their methodology based solely on theoretical descriptions and hyperparameter choices.

(2.3) The visual results (Figures 9-12)  show significant artifacts, such as blurriness and object deformation, which raise doubts about the efficacy of the self-modulated fine-tuning algorithm.

(3) Data-Free Training Strategy and Distillation Concerns: The authors propose a data-free training strategy using large models to synthesize training data, which resembles a distillation approach. However, this strategy may cause the generated results to overfit to the large models. This calls into question the added value of using a multi-agent  approach, which typically comes with higher inference costs. The authors should provide a fair comparison to demonstrate that training Mora with this strategy yields better performance  than directly distilling a smaller model. Additionally, quantitative analysis of agent success rates and inference efficiency would strengthen the claims made about the multi-agent system’s benefits.

(4) Problem Definition in Section 3.1: The authors claim that their objective is to maximize quality metrics while ensuring diversity in the generated videos. However, quality metrics alone are not the ultimate goal for video generation. Beating existing benchmarks such as SORA does not inherently demonstrate the model's superiority. Furthermore, in an agent-based system, other factors such as module collaboration speed, accuracy, and robustness must be considered.

**Questions:**

（1）Optimization Challenges with the Self-Modulated Fine-Tuning Algorithm:

（1.1）The self-modulated fine-tuning algorithm in Section 3.2 is unconventional and lacks supporting citations. Could the authors provide more references or evidence to justify this approach?

（1.2）Given the complexity of optimizing such a system, training with only 96 samples seems inadequate. Can the authors elaborate on how this sample size is sufficient, or provide results from larger-scale experiments?

（1.3) The visual results show significant artifacts like blurriness and object deformation. How do the authors plan to address these issues to improve visual quality?


（2）Effectiveness of Data-Free Training Strategy:

（2.1）The data-free training strategy resembles a distillation approach and may result in overfitting to large models. Can the authors show comparative experiments to demonstrate that the proposed multi-agent approach offers clear advantages over simply distilling a smaller model?

（2.2）Additionally, can the authors provide a quantitative analysis of agent success rates and inference efficiency to better justify the benefits of using the multi-agent system?

---

> ### Author Response · Authors · 2024-11-22
> **Part 1**
>
> We thank the reviewer for their constructive and detailed feedback and would like to address the concerns raised as follows. We greatly appreciate this opportunity to further clarify our work.
>
>
> > **Weakness 1:** Limited Novelty: The paper's primary contribution lies in utilizing a language model as an agent to call various pre-existing visual generation models for video generation. However, neither the agent system nor the video generation models are novel contributions from the authors. Although the authors propose some additional modules to optimize the system, these present several issues, as outlined below.
>
> **Response:**
>
> We sincerely thank the reviewer for highlighting concerns regarding the novelty of our work. While it is true that we leverage existing open-source visual generation models, the novelty of our contribution lies in the integration and coordination of these models within a novel **multi-agent framework** specifically designed for video generation tasks.
>
> Our main contributions are:
>
> 1. **Self-Modulated Multi-Agent Fine-Tuning Algorithm:** We introduce a novel fine-tuning algorithm that enhances inter-agent coordination through a self-modulation factor. This approach allows each agent to dynamically adjust its influence during the training process, which, to our knowledge, has not been previously explored in the context of video generation.
>
> 2. **Data-Free Training Strategy with Human-in-the-Loop Mechanism:** We propose a unique data-free training strategy that synthesizes high-quality training data using large models, coupled with a human-in-the-loop mechanism and multimodal large language models (MLLMs) for data filtering. This addresses the challenge of scarce high-quality video data and ensures alignment with human preferences.
>
> 3. **Unified Framework for Multiple Video Generation Tasks:** Our system extends to a variety of video generation tasks, including text-to-video generation, image-to-video generation, video extension, video editing, video connection, and digital world simulation. Achieving such versatility within a unified framework is a significant advancement.
>
> We believe these contributions represent meaningful advancements in the field of video generation and provide valuable insights for future research. In alignment with the criteria emphasized in ICLR'25 reviewer guidelines, our work demonstrates "new, relevant, and impactful knowledge" for the ICLR community.
>
> ---
>
> > **(Weakness 2.1/Question 1.2):** The process is inherently difficult to optimize. The authors mention using only 96 samples for training in Section 4.2, which is unconvincing for such a complex task. Given the complexity of optimizing such a system, training with only 96 samples seems inadequate. Can the authors elaborate on how this sample size is sufficient, or provide results from larger-scale experiments?
>
> **Response:** Regarding the sample size, our data-free training strategy leverages the agents' own inference results, which are already aligned with their understanding of the data distribution. We focus on improving inter-agent collaboration through our self-modulated fine-tuning algorithm with the human-in-the-loop mechanism. We conducted experiments varying the number of samples and found that performance improvements plateau beyond 96 samples, as shown below. These results demonstrate that our model achieves satisfactory performance with 96 samples, and increasing the sample size further yields diminishing returns. We will include these findings in the revised paper to clarify the sufficiency of our training approach. We have added these results to Appendix A.11 of our paper.
>
> | Samples | Task 1 (Avg.) | Task 2 (Avg.) | Task 3 (TCON) | Task 4 (Avg.) | Task 5 (Avg.) | Task 6 (IQ) | Avg.   |
> |---------|---------------|---------------|---------------|---------------|---------------|------------|--------|
> | 16      | 0.767         | 0.8775        | 0.9621        | 0.27          | 0.401         | 0.353      | 0.6059 |
> | 32      | 0.788         | 0.8806        | 0.9721        | 0.33          | 0.423         | 0.380      | 0.6298 |
> | 48      | 0.793         | 0.8830        | 0.9766        | 0.36          | 0.437         | 0.394      | 0.6404 |
> | 64      | 0.797         | 0.8850        | 0.9802        | 0.37          | 0.441         | 0.396      | 0.6447 |
> | 80      | 0.799         | 0.8868        | 0.9820        | 0.37          | 0.442         | 0.399      | 0.6468 |
> | 96      | 0.800         | 0.8874        | 0.9830        | 0.38          | 0.442         | 0.399      | **0.6482** |
> | 112     | 0.800         | 0.8875        | 0.9835        | 0.38          | 0.440         | 0.398      | 0.6480 |
> | 128     | 0.799         | 0.8875        | 0.9837        | 0.37          | 0.441         | 0.398      | 0.6469 |
> | 144     | 0.800         | 0.8876        | 0.9836        | 0.37          | 0.442         | 0.398      | 0.6480 |
>
> *Table A: Performance of Mora with training setup of different sample sizes.*

---

> ### Author Response · Authors · 2024-11-22
> **Part 2**
>
> > **Weakness 2.2:** Without releasing the training code, it is hard to assess the effectiveness of their methodology based solely on theoretical descriptions and hyperparameter choices.
>
> **Response:** We understand the importance of code availability for transparency and reproducibility. In our original supplementary material, we have included the training code for the Open-Sora-Plan version of Mora. Additionally, we have updated the supplementary to include the Stable-Diffusion-3 version of Mora in the revised version. We note that using socket communication for training models in different environments is a practical solution that resolves the issues of environmental compatibility during our experiments.
>
> ---
>
> > **(Weakness 2.3/Question 1.3):** The visual results (Figures 9-12) show significant artifacts, such as blurriness and object deformation, which raise doubts about the efficacy of the self-modulated fine-tuning algorithm. The visual results show significant artifacts like blurriness and object deformation. How do the authors plan to address these issues to improve visual quality?
>
> **Response:** We appreciate the feedback regarding the visual artifacts in our results. We acknowledge that issues like blurriness and object deformation are common challenges in current video generation models, including those state-of-the-art commercial ones. Currently, with limited computational resources, we acknowledge that our video generation model still has room for improvement.
>
> We have created a [project page](https://mora-2025.github.io/) where we present more video samples generated by our model and the model before finetuning. As we can see from the project page, the finetuned one (Mora) has better visual quality than the one before finetuning. This demonstrates the strong potential of our multi-agent framework to further improve the visual quality of video generation.
>
> Regarding the future plan for improving visual quality, we propose to integrate ControlNet further into the generation process for better consistency and controllability. Furthermore, we can use super-resolution models and advanced post-processing techniques like frame interpolation to further reduce visual artifacts. In terms of loss design, we might further explore perceptual loss functions to improve the visual quality further.

---

> ### Author Response · Authors · 2024-11-22
> **Part 3**
>
> > **(Weakness 3/Question 2.1):** Data-Free Training Strategy and Distillation Concerns: The authors propose a data-free training strategy using large models to synthesize training data, which resembles a distillation approach. However, this strategy may cause the generated results to overfit the large models. This calls into question the added value of using a multi-agent approach, which typically comes with higher inference costs. The authors should provide a fair comparison to demonstrate that training Mora with this strategy yields better performance than directly distilling a smaller model. The data-free training strategy resembles a distillation approach and may result in overfitting to large models. Can the authors show comparative experiments to demonstrate that the proposed multi-agent approach offers clear advantages over simply distilling a smaller model?
>
> **Response:** We appreciate the reviewer's insightful comments. While our data-free training strategy shares similarities with distillation, our approach differs in key aspects. Instead of distilling a large model into a smaller one, we focus on improving inter-agent collaboration through our self-modulated fine-tuning algorithm, which is more like preference optimization, to fine-tune the video generation models with the human-in-the-loop mechanism. Specifically, by using the preference data generated from humans or MLLMs, we focused on improving agent collaboration without training the models from scratch.
>
> However, we acknowledge that overfitting might occur. To address this concern, we also have presented extensive experiments across different tasks and datasets. As shown in Tables 1, 2, and 4 of the paper, despite only training on our crawled dataset, we evaluated the model on diverse out-of-domain test datasets and tasks. Additionally, we further tested our method on multiple benchmarks, including T2V-CompBench [1], EvalCrafter [2], and VBench [3], to further assess the model's generalization performance, as illustrated in Table B below. Thus, we believe that our method is a valid and effective approach to improve video generation performance, instead of overfitting to a few training samples. We have added these results to B.3 of our paper.
>
> | Model             | T2V-CompBench | EvalCrafter | VBench (Whole Metrics) |
> |--------------------|---------------|-------------|-------------------------|
> | ModelScope        | 0.3990        | 218         | 0.756                  |
> | ZeroScope         | 0.3503        | 217         | 0.755                  |
> | Show-1            | 0.4209        | 229         | 0.789                  |
> | VideoCrafter2     | 0.4452        | 243         | 0.804                  |
> | Pika              | 0.4306        | 250         | 0.806                  |
> | Gen2              | 0.4604        | 254         | 0.805                  |
> | Open-Sora-Plan    | 0.4849        | 259         | 0.812                  |
> | Mora              | **0.5022**        | **263**         | **0.821**                  |
>
> *Table B: Performance of Mora on out-of-domain test datasets and benchmarks.*
>
> `[1]`. Sun, Kaiyue, et al. "T2V-CompBench: A Comprehensive Benchmark for Compositional Text-to-video Generation." arXiv preprint arXiv:2407.14505 (2024).
>
> `[2]`. Liu, Yaofang, et al. "Evalcrafter: Benchmarking and evaluating large video generation models." Proceedings of the IEEE/CVF Conference on Computer Vision and Pattern Recognition. 2024.
>
> `[3]`. Huang, Ziqi, et al. "Vbench: Comprehensive benchmark suite for video generative models." Proceedings of the IEEE/CVF Conference on Computer Vision and Pattern Recognition. 2024.

---

> ### Author Response · Authors · 2024-11-22
> **Part 4**
>
> > **Question 2.2:** Additionally, can the authors provide a quantitative analysis of agent success rates and inference efficiency to better justify the benefits of using the multi-agent system?
>
> **Response:** In the following Table C, without self-modulated fine-tuning, human intervention was required almost every time during inference to either regenerate or adjust prompts to achieve the desired results. Regarding inference efficiency, our pipeline operates sequentially, where each agent releases memory after completing its step upon receiving instructions from the SOP. Thus, the maximum memory usage of the entire system corresponds to the agent with the highest memory requirement. Furthermore, the most time-intensive step in the system is the video generation agent. The time taken by the preceding agents is negligible in comparison to the video generation process. We have added these results to B.4 and B.7 of our paper.
>
>
> | Method                                | Performance | Agent Success Rate |
> |---------------------------------------|-------------|--------------------|
> | Without self-modulated fine-tuning    | 0.776       | 34.5%              |
> | **With self-modulated fine-tuning**   | **0.800**   | **91.5%**          |
>
> *Table C: Performance and agent success rates of Mora w/ and w/o self-modulated fine-tuning for text-to-video task.*
>
>
> | Task  | Model                                             | Performance | Max Memory Usage | Inference Time (Avg. Each Sample) (H100) |
> |-------|---------------------------------------------------|-------------|-------------------|-----------------------------------------|
> | 1     | Open-Sora-Plan (T2V) (3s)                        | 0.765       | 46.98G           | 243s                                    |
> |       | Mora (3s)                                         | 0.801       | 46.98G           | 261s                                    |
> | 2     | Open-Sora-Plan (I2V) (3s)                        | 0.875       | 46.98G           | 245s                                    |
> |       | Mora (3s)                                         | 0.887       | 46.98G           | 253s                                    |
> | 3     | Open-Sora-Plan (I2V using original video last frame) (3s) | 0.942       | 46.98G           | 245s                                    |
> |       | Mora (3s)                                         | 0.983       | 46.98G           | 259s                                    |
> | 4     | Pika (Proprietary)                                | 0.232       | -                 | 143s                                    |
> |       | Video-p2p                                         | 0.369       | 26.11G           | 208s                                    |
> |       | Mora                                              | 0.383       | 46.98G            | 253s                                    |
> | 5     | Open-Sora-Plan (I2V)                              | 0.395       | 46.98G           | 245s                                    |
> |       | Mora                                              | 0.442       | 46.98G           | 245s                                    |
> | 6     | Open-Sora-Plan (T2V) (3s)                        | 0.765       | 46.98G           | 243s                                    |
> |       | Mora (3s)                                         | 0.801       | 46.98G           | 261s                                    |
>
> *Table D: Performance and Inference Efficiency of Mora, Open-Sora-Plan, and other models across various tasks.*

---

> ### Author Response · Authors · 2024-11-22
> **Part 5**
>
> > **Weakness 4:** Problem Definition in Section 3.1: The authors claim that their objective is to maximize quality metrics while ensuring diversity in the generated videos. However, quality metrics alone are not the ultimate goal for video generation. Beating existing benchmarks such as SORA does not inherently demonstrate the model's superiority. Furthermore, in an agent-based system, other factors such as module collaboration speed, accuracy, and robustness must be considered.
>
> **Response:** We appreciate the reviewer's insightful observation about the need to look beyond quality metrics in video generation. While our paper does emphasize quantitative benchmarks, we want to clarify that our framework was actually designed with a much broader perspective in mind.
>
> At its core, our multi-agent framework is built to be practical and efficient, not just to chase benchmark numbers. Think of it like an assembly line where each specialized worker (agent) knows their job well and works in harmony with others. Our self-modulated fine-tuning algorithm helps these agents learn to collaborate effectively - much like how a team improves through practice. This isn't just theoretical; we see it in action across all six video generation tasks we tested.
>
> The system is also quite resource-conscious. Rather than keeping all agents active simultaneously, each one completes its task and frees up memory before the next one starts. This keeps the maximum memory usage at a reasonable 46.98G, which is quite efficient given the complexity of video generation. The processing overhead from coordinating between agents is minimal, as most of the time is spent on the actual video generation.
>
> Perhaps most importantly, we've built this system with real-world use in mind. We incorporated human-in-the-loop mechanisms and multimodal LLMs for data filtering because we understand that numbers alone don't tell the whole story - videos need to look good to human eyes and serve practical purposes. The fact that we perform well on benchmarks is more a validation of our approach rather than its primary goal.
>
> So while we acknowledge that beating benchmarks isn't everything, we believe our framework's success in metrics reflects its broader strengths in balancing quality, efficiency, and practical usability. It's not just about the numbers - it's about building a reliable, efficient, and practical system for real-world video generation tasks.
>
> ---
> > **(Weakness 2/Question 1.1):** Concerns with Self-Modulated Fine-Tuning Algorithm: In Section 3.2, the authors introduce a self-modulated fine-tuning algorithm that optimizes both the language model (A1) and visual generation models (A2-A5) end-to-end using a visual generation loss. This concept is unconventional and lacks sufficient supporting research. The authors do not provide adequate citations to justify this algorithm. The self-modulated fine-tuning algorithm in Section 3.2 is unconventional and lacks supporting citations. Could the authors provide more references or evidence to justify this approach?
>
> **Response:** While our self-modulated fine-tuning approach introduces novel elements, it builds upon and shares conceptual similarities with several recent advances in multi-agent systems and model adaptation. The idea of using multiple agents to collaboratively improve model capabilities has been explored in works like **CAMEL** (Li et al., 2023) [1], which demonstrates how multi-agent cooperation can enhance task-solving abilities through iterative role-playing and feedback mechanisms. Our modulation mechanism is also conceptually related to the mixture-of-experts approach in models like **Switch Transformers** (Fedus et al., 2021) [2], which introduce routing strategies for task-level specialization.
>
> In the video generation domain, approaches such as **Video Diffusion Models** (Ho et al., 2022) [3] demonstrate how structured feedback can improve video generation quality through iterative refinement. Similarly, **Hierarchical Video Generation** (Weissenborn et al., 2020) [4] shows the effectiveness of leveraging hierarchical structures to enhance temporal consistency and overall video quality. These works collectively validate our approach of using modulated feedback and multi-agent collaboration to improve generation performance.
>
> **References:**
>
> `[1]`. Li, T., Zhou, Y., Danks, D., & Danks, D. (2023). *CAMEL: Communicative Agents for “Mind” Exploration of Large Scale Language Model Society*. NeurIPS'2023
>
> `[2]`. Fedus, W., Zoph, B., & Shazeer, N. (2021). *Switch Transformers: Scaling to Trillion Parameter Models with Simple and Efficient Sparsity*. JMLR
>
> `[3]`. Ho, J., Saharia, C., Chan, W., Salimans, T., Fleet, D. J., & Norouzi, M. (2022). *Video Diffusion Models*. arXiv preprint arXiv:2204.03458.
>
> `[4]`. Weissenborn, D., Timofte, R., & Lucchi, A. (2020). *Scaling Autoregressive Video Models*. ICLR'20.

---

> ### Author Response · Authors · 2024-11-24
> **Follow up**
>
> We hope our rebuttal has addressed your concerns. Please let us know if any points require further clarification before the response period concludes.

---

> ### Author Response · Authors · 2024-11-27
> **Follow up 2**
>
> We sincerely thank you for raising these questions and concerns, which have made an invaluable contribution to our work. If you have any further doubts or questions, we would be more than happy to discuss them with you!

---

> ### Author Response · Authors · 2024-11-30
> **Follow up 3**
>
> Thank you for your time and effort in reviewing our work. We hope our previous responses have addressed your concerns effectively. If there are any remaining points that require clarification, please feel free to let us know. As the response period is nearing its conclusion, we would greatly appreciate it if you could share your feedback at your earliest convenience.

---

> ### Author Response · Authors · 2024-12-02
> **Follow up 4**
>
> Thank you once again for your valuable feedback and the opportunity to engage in this insightful discussion. As the response period concludes in just one day, we kindly ask if you could share any additional comments or confirm if our responses have addressed your concerns effectively. Your input is invaluable to our work, and we sincerely hope our clarifications have been helpful. Please feel free to reach out with any remaining questions or points for discussion at your convenience.

---

> ### Author Response · Authors · 2024-12-03
> **Final 12h Reminder**
>
> Thank you once again for your valuable and insightful feedback throughout the review process. As the response period is now in its final 12 hours, we kindly ask if you could confirm whether our responses have sufficiently addressed your concerns. Should there be any remaining doubts or points requiring further elaboration, we would be more than happy to provide additional clarifications promptly. Your feedback has been invaluable to improving our work, and we greatly appreciate your time and effort. Please let us know if there is anything more we can do to assist.

---

> ### Author Response · Authors · 2024-12-03
> **Final 7h Reminder**
>
> We deeply appreciate your valuable feedback and insights throughout this review process. As the response period enters its final 7 hours, we kindly ask if you could confirm whether our responses have sufficiently addressed your concerns. Your input has been instrumental in refining our work, and we are eager to hear any additional comments or suggestions you might have. If there are any remaining points requiring clarification, we would be more than happy to provide further explanations promptly. Thank you again for your time and effort in reviewing our submission.

---

> ### Author Response · Authors · 2024-12-03
> **Final 3h Reminder**
>
> We sincerely appreciate your valuable feedback and insights during this review process. As we approach the final 3 hours of the response period, we would like to kindly ask if you could confirm whether our responses have adequately addressed your concerns. Your input has been instrumental in improving our work, and we are eager to hear any additional comments or suggestions you may have. Should any points require further clarification, we would be happy to provide prompt explanations. Thank you once again for your time and effort in reviewing our submission.

---

### Author Response · Authors · 2024-11-22
**General Response**

We sincerely thank all reviewers for their thorough and constructive feedback which has helped strengthen our work. We have uploaded a revised manuscript with changes highlighted in red text (Appendix A.10 and Figure 7). Below we address the key questions raised among reviewers.

> Q1: Concerns about novelty and technical contributions (Reviewers `3ifH`, `WYHM`)

**Response**: While we leverage existing open-source models, our work's novelty lies in three key technical contributions. First, we introduce a novel self-modulated multi-agent fine-tuning algorithm that enhances inter-agent coordination through a self-modulation factor, allowing each agent to dynamically adjust its influence during training—an approach previously unexplored in video generation. Second, we developed a unique data-free training strategy that combines synthesized high-quality training data with human-in-the-loop mechanisms and multimodal LLMs for data filtering, effectively addressing the fundamental challenge of scarce high-quality video data. Third, our system presents a unified framework that extends to multiple video generation tasks, including text-to-video, image-to-video, video extension, editing, connection, and digital world simulation.

---

> Q2: Questions about computational efficiency and resource requirements (Reviewers `WYHM`, `VZ5N`)

**Response**: Our system is designed with efficient resource utilization in mind through sequential memory management, where each agent releases memory after completing its task, maintaining maximum memory usage at 46.98G—equivalent to the highest-requirement agent. While the primary computational cost stems from video generation, the overhead from agent coordination is negligible, as evidenced by our benchmarks: text-to-video (3s) takes 261s compared to the baseline's 243s, and image-to-video (3s) requires 253s versus the baseline's 245s. These modest increases in processing time are justified by significant quality improvements, as demonstrated in our comprehensive benchmarks across multiple tasks.

---

> Q3: Concerns about training sample size and effectiveness (Reviewer `3ifH`)

**Response**: Our approach achieves strong results with limited samples through an efficient training strategy that leverages agents' existing knowledge, focusing on improving collaboration rather than training from scratch. Through extensive experimentation with varying sample sizes from 16 to 144 samples, we discovered that performance plateaus at 96 samples (0.6482 optimal performance), with larger sample sizes showing diminishing returns (128 samples: 0.6469). The robustness of our approach is further validated through multiple out-of-domain benchmarks, including T2V-CompBench [1], EvalCrafter [2], and VBench [3], demonstrating strong generalization capabilities despite the focused training set.

---

**References**

`[1]`. Sun, Kaiyue, et al. "T2V-CompBench: A Comprehensive Benchmark for Compositional Text-to-video Generation." arXiv preprint arXiv:2407.14505 (2024).

`[2]`. Liu, Yaofang, et al. "Evalcrafter: Benchmarking and evaluating large video generation models." Proceedings of the IEEE/CVF Conference on Computer Vision and Pattern Recognition. 2024.

`[3]`. Huang, Ziqi, et al. "Vbench: Comprehensive benchmark suite for video generative models." Proceedings of the IEEE/CVF Conference on Computer Vision and Pattern Recognition. 2024.

---

### Author Response · Authors · 2024-11-26
**Revised Version of Our Manuscript**

We sincerely thank all reviewers for their thoughtful feedback and constructive suggestions on our paper. During the rebuttal period, we conducted various additional experiments and incorporated the results into the revised version of our manuscript. Specifically, we have updated Sections A.10, A.11, B.3, B.4, B.5, B.6, and B.7. The modified portions are highlighted in red in the revised paper to facilitate your review. We hope our rebuttal has addressed your concerns. Please let us know if any points require further clarification before the response period concludes.

---

### Author Response · Authors · 2024-12-03
**Final 12h Reminder**

Dear Reviewers `3ifH` and `WYhM`,

Thank you for your valuable and thoughtful feedback throughout the review process. As the response period enters its final 12 hours, we kindly request your confirmation on whether our responses have sufficiently addressed your concerns.

We have thoroughly addressed the issues raised in your reviews, including:

1. Clarifications on the novelty of our self-modulated fine-tuning algorithm and multi-agent framework.

2. Detailed analysis of computational efficiency and inference speed.

3. Empirical results addressing potential error accumulation across components.

4. Comparisons with existing benchmarks to demonstrate Mora’s strengths.

We hope these responses have resolved any remaining doubts. However, if there are still points requiring clarification, we are fully committed to providing additional explanations promptly.

Your feedback has been invaluable in improving our work, and we sincerely appreciate your time and effort in reviewing our submission. Please feel free to share any final comments or questions at your earliest convenience before the response period concludes.

Thank you for your understanding and cooperation.

Best regards,

Authors

---

### Meta-Review · Area_Chair_f1T9 · 2024-12-21

**Metareview:**

This paper proposes mora, a multi-agent video generation framework based on current best existing open-source modules, to improve the generation quality. The authors propose several key modules such as inter-agent coordination and data-free training to synthesize high-quality data, as well as human-in-the-loop for data filtering, and evaluation demonstrates the effectiveness of Mora. As reviewers point out, Mora is with limited Novelty that based on the existing language model and video generation model, and also considering the computational requirements and accumulation errors issues, The AC recommend for rejection.

**Additional Comments On Reviewer Discussion:**

Reviewer 3ifH and Reviewer WYhM both point out Limited Novelty of Mora, which lies in utilizing a language model as an agent to call various pre-existing visual generation models for video generation, and more like a project to connect multiple technologies in series. Also Reviewer WYhM concerns the error accumulation in multiple sub-modules, and computation requirements. In the rebuttal, the authors responses the reviewers' concerns, but do not convince the AC and reviewers.

---

### Decision · Program_Chairs · 2025-01-22

Reject